# ON THE LIMITATIONS OF MULTIMODAL VAEs

**Imant Daunhawer, Thomas M. Sutter, Kieran Chin-Cheong, Emanuele Palumbo & Julia E. Vogt**
Department of Computer Science
ETH Zurich
`dimant@ethz.ch`

## ABSTRACT

Multimodal variational autoencoders (VAEs) have shown promise as efficient generative models for weakly-supervised data. Yet, despite their advantage of weak supervision, they exhibit a gap in generative quality compared to unimodal VAEs, which are completely unsupervised. In an attempt to explain this gap, we uncover a fundamental limitation that applies to a large family of mixture-based multimodal VAEs. We prove that the sub-sampling of modalities enforces an undesirable upper bound on the multimodal ELBO and thereby limits the generative quality of the respective models. Empirically, we showcase the generative quality gap on both synthetic and real data and present the tradeoffs between different variants of multimodal VAEs. We find that none of the existing approaches fulfills all desired criteria of an effective multimodal generative model when applied on more complex datasets than those used in previous benchmarks. In summary, we identify, formalize, and validate fundamental limitations of VAE-based approaches for modeling weakly-supervised data and discuss implications for real-world applications.

## 1 INTRODUCTION

In recent years, multimodal VAEs have shown great potential as efficient generative models for weakly-supervised data, such as pairs of images or paired images and captions. Previous works (Wu and Goodman, 2018; Shi et al., 2019; Sutter et al., 2020) demonstrate that multimodal VAEs leverage weak supervision to learn generalizable representations, useful for downstream tasks (Dorent et al., 2019; Minoura et al., 2021) and for the conditional generation of missing modalities (Lee and van der Schaar, 2021). However, despite the advantage of weak supervision, state-of-the-art multimodal VAEs consistently underperform when compared to simple unimodal VAEs in terms of generative quality.[1] This paradox serves as a starting point for our work, which aims to explain the observed lack of generative quality in terms of a fundamental limitation that underlies existing multimodal VAEs.

What is limiting the generative quality of multimodal VAEs? We find that the sub-sampling of modalities during training leads to a problem that affects all *mixture-based* multimodal VAEs—a family of models that subsumes the MMVAE (Shi et al., 2019), MoPoE-VAE (Sutter et al., 2021), and a special case of the MVAE (Wu and Goodman, 2018). We prove that modality sub-sampling enforces an undesirable upper bound on the multimodal ELBO and thus prevents a tight approximation of the joint distribution when there is modality-specific variation in the data. Our experiments demonstrate that modality sub-sampling can explain the gap in generative quality compared to unimodal VAEs and that the gap typically increases with each additional modality. Through extensive ablations on three different datasets, we validate the generative quality gap between unimodal and multimodal VAEs and present the tradeoffs between different approaches.

Our results raise serious concerns about the utility of multimodal VAEs for real-world applications. We show that none of the existing approaches fulfills all desired criteria (Shi et al., 2019; Sutter et al., 2020) of an effective multimodal generative model when applied to slightly more complex datasets than used in previous benchmarks. In particular, we demonstrate that generative coherence (Shi et al., 2019) cannot be guaranteed for any of the existing approaches, if the information shared between modalities cannot be predicted in expectation across modalities. Our findings are particularly relevant for applications on datasets with a relatively high degree of modality-specific variation, which is a typical characteristic of many real-world datasets (Baltrušaitis et al., 2019).

---

[1]The lack of generative quality can even be recognized by visual inspection of the qualitative results from previous works; for instance, see the supplementaries of Sutter et al. (2021) or Shi et al. (2021).

## 2 RELATED WORK

First, to put multimodal VAEs into context, let us point out that there is a long line of research focused on learning multimodal generative models based on a wide variety of methods. There are several notable generative models with applications on pairs of modalities (e.g., Ngiam et al., 2011; Srivastava and Salakhutdinov, 2014; Wu and Goodman, 2019; Lin et al., 2021; Ramesh et al., 2021), as well as for the specialized task of image-to-image translation (e.g., Huang et al., 2018; Choi et al., 2018; Liu et al., 2019). Moreover, generative models can use labels as side information (Ilse et al., 2019; Tsai et al., 2019; Wieser et al., 2020); for example, to guide the disentanglement of shared and modality-specific information (Tsai et al., 2019). In contrast, multimodal VAEs do not require strong supervision and can handle a large and variable number of modalities efficiently. They learn a joint distribution over multiple modalities, but also enable the inference of latent representations, as well as the conditional generation of missing modalities, given any subset of modalities (Wu and Goodman, 2018; Shi et al., 2019; Sutter et al., 2021).

Multimodal VAEs are an extension of VAEs (Kingma and Welling, 2014) and they belong to the class of multimodal generative models with encoder-decoder architectures (Baltrušaitis et al., 2019). The first multimodal extensions of VAEs (Suzuki et al., 2016; Hsu and Glass, 2018; Vedantam et al., 2018) use separate inference networks for every subset of modalities, which quickly becomes intractable as the number of inference networks required grows exponentially with the number of modalities. Starting with the seminal work of Wu and Goodman (2018), multimodal VAEs were developed as an *efficient* method for multimodal learning. In particular, multimodal VAEs enable the inference of latent representations, as well as the conditional generation of missing modalities, given any subset of input modalities. Different types of multimodal VAEs were devised by decomposing the joint encoder as a product (Wu and Goodman, 2018), mixture (Shi et al., 2019), or mixture of products (Sutter et al., 2021) of unimodal encoders respectively. A commonality between these approaches is the sub-sampling of modalities during training—a property we will use to define the family of *mixture-based* multimodal VAEs. For the MMVAE and MoPoE-VAE, the sub-sampling is a direct consequence of defining the joint encoder as a mixture distribution over different subsets of modalities. Further, our analysis includes a special case of the MVAE *without* ELBO sub-sampling, which can be seen as another member of the family of mixture-based multimodal VAEs (Sutter et al., 2021). The MVAE was originally proposed with "ELBO sub-sampling", an additional training paradigm that was later found to result in an incorrect bound on the joint distribution (Wu and Goodman, 2019). While this training paradigm is also based on the sub-sampling of modalities, the objective differs from mixture-based multimodal VAEs in that the MVAE does not reconstruct the missing modalities from the set of sub-sampled modalities.[2]

Table 1 provides an overview of the different variants of mixture-based multimodal VAEs and the properties that one can infer from empirical results in previous works (Shi et al., 2019; 2021; Sutter et al., 2021). Most importantly, there appears to be a tradeoff between generative quality and generative coherence (i.e., the ability to generate semantically related samples across modalities). Our work explains *why* the generative quality is worse for models that sub-sample modalities (Section 4) and shows that a tighter approximation of the joint distribution can be achieved without sub-sampling (Section 4.3). Through systematic ablations, we validate the proposed theoretical limitations and showcase the tradeoff between generative quality and generative coherence (Section 5.1). Our experiments also reveal that generative coherence cannot be guaranteed for more complex datasets than those used in previous benchmarks (Section 5.2).

## 3 MULTIMODAL VAEs, IN DIFFERENT FLAVORS

Let $\boldsymbol{X} := \{X_1, \ldots, X_M\}$ be a set of random vectors describing $M$ modalities and let $\boldsymbol{x} := \{\boldsymbol{x}_1, \ldots, \boldsymbol{x}_M\}$ be a sample from the joint distribution $p(\boldsymbol{x}_1, \ldots, \boldsymbol{x}_M)$. For conciseness, denote subsets of modalities by subscripts; for example, $\boldsymbol{X}_{\{1,3\}}$ or $\boldsymbol{x}_{\{1,3\}}$ respectively for modalities 1 and 3.

Throughout this work, we assume that all modalities are described by discrete random vectors (e.g., pixel values), so that we can assume non-negative entropy and conditional entropy terms. Definitions for all required information-theoretic quantities are provided in Appendix A.

---

[2]For completeness, in Appendix C, we also analyze the effect of ELBO sub-sampling.

Table 1: Overview of multimodal VAEs. Entries for generative quality and generative coherence denote properties that were observed empirically in previous works. The lightning symbol ($\natural$) denotes properties for which our work presents contrary evidence. This overview abstracts technical details, such as importance sampling and ELBO sub-sampling, which we address in Appendix C.

| Model | Decomposition of $p_\theta(z \mid x)$ | Modality sub-sampling | Generative quality | Generative coherence |
|---|---|---|---|---|
| **MVAE** (Wu and Goodman, 2018) | $\prod_{i=1}^{M} p_\theta(z \mid x_i)$ | ✗ | good | poor |
| **MMVAE** (Shi et al., 2019) | $\frac{1}{M} \sum_{i=1}^{M} p_\theta(z \mid x_i)$ | ✓ | limited | good $\natural$ |
| **MoPoE-VAE** (Sutter et al., 2021) | $\frac{1}{|\mathcal{P}(M)|} \sum_{A \in \mathcal{P}(M)} \prod_{i \in A} p_\theta(z \mid x_i)$ | ✓ | limited | good $\natural$ |

## 3.1 THE MULTIMODAL ELBO

**Definition 1.** *Let $p_\theta(z \mid x)$ be a stochastic encoder, parameterized by $\theta$, that takes multiple modalities as input. Let $q_\phi(x \mid z)$ be a variational decoder (for all modalities), parameterized by $\phi$, and let $q(z)$ be a prior. The multimodal evidence lower bound (ELBO) on $\mathbb{E}_{p(x)}[\log p(x)]$ is defined as*

$$\mathcal{L}(x; \theta, \phi) \coloneqq \mathbb{E}_{p(x)p_\theta(z \mid x)}[\log q_\phi(x \mid z)] - \mathbb{E}_{p(x)}[D_{KL}(p_\theta(z \mid x) \,\|\, q(z))] . \tag{1}$$

The multimodal ELBO (Definition 1), first introduced by Wu and Goodman (2018), is the objective maximized by all multimodal VAEs and it forms a variational lower bound on the expected log-evidence.[3] The first term denotes the estimated log-likelihood of all modalities and the second term is the KL-divergence between the stochastic encoder and the prior. We take an information-theoretic perspective using the variational information bottleneck (VIB) from Alemi et al. (2017) and employ the standard notation used in multiple previous works (Alemi et al., 2017; Poole et al., 2019). Similar to the latent variable model approach, the VIB derives the ELBO as a variational lower bound on the expected log-evidence, but, in addition, the VIB is a more general framework for optimization that allows us to reason about the underlying information-theoretic quantities of interest (for details on the VIB and its notation, please see Appendix B.1).

Note that the above definition of the multimodal ELBO requires that the complete set of modalities is available. To overcome this limitation and to learn the inference networks for different subsets of modalities, existing models use different *decompositions* of the joint encoder, as summarized in Table 1. Recent work shows that existing models can be generalized by formulating the joint encoder as a mixture of products of experts (Sutter et al., 2021). Analogously, in the following, we generalize existing models to define the family of mixture-based multimodal VAEs.

## 3.2 THE FAMILY OF MIXTURE-BASED MULTIMODAL VAEs

Now we introduce the family of mixture-based multimodal VAEs, which subsumes the MMVAE, MoPoE-VAE, and a special case of the MVAE without ELBO sub-sampling. We first define an encoder that generalizes the decompositions used by existing models:

**Definition 2.** *Let $\mathcal{S} = \{(A, \omega_A) \mid A \subseteq \{1, \ldots, M\}, A \neq \emptyset, \omega_A \in [0, 1]\}$ be an arbitrary set of non-empty subsets $A$ of modalities and corresponding mixture coefficients $\omega_A$, such that $\sum_{A \in \mathcal{S}} \omega_A = 1$. Define the stochastic encoder to be a mixture distribution: $p_\theta^{\mathcal{S}}(z \mid x) \coloneqq \sum_{A \in \mathcal{S}} \omega_A \, p_\theta(z \mid x_A)$.*

In the above definition and throughout this work, we write $A \in \mathcal{S}$ to abbreviate $(A, \omega_A) \in \mathcal{S}$. To define the family of mixture-based multimodal VAEs, we restrict the family of models optimizing the multimodal ELBO to the subfamily of models that use a mixture-based stochastic encoder.

**Definition 3.** *The family of mixture-based multimodal VAEs is comprised of all models that maximize the multimodal ELBO using a stochastic encoder $p_\theta^{\mathcal{S}}(z \mid x)$ that is consistent with Definition 2. In particular, we define the family in terms of all models that maximize the following objective:*

$$\mathcal{L}_{\mathcal{S}}(x; \theta, \phi) = \sum_{A \in \mathcal{S}} \omega_A \left\{ \mathbb{E}_{p(x)p_\theta(z \mid x_A)}[\log q_\phi(x \mid z)] - \mathbb{E}_{p(x)} \left[ D_{KL} \left( p_\theta(z \mid x_A) \,\|\, q(z) \right) \right] \right\} . \tag{2}$$

---

[3]Even though we write the expectation over $p(x)$, for the estimation of the ELBO we still assume that we only have access to a finite sample from the training distribution $p(x)$. The notation is used for consistency with the well-established information-theoretic perspective on VAEs (Alemi et al., 2017; Poole et al., 2019).

In Appendix B.2, we show that the objective $\mathcal{L}_{\mathcal{S}}(\boldsymbol{x}; \theta, \phi)$ is a lower bound on $\mathcal{L}(\boldsymbol{x}; \theta, \phi)$ (which makes it an ELBO) and explain how, for different choices of the set of subsets $\mathcal{S}$, the objective $\mathcal{L}_{\mathcal{S}}(\boldsymbol{x}; \theta, \phi)$ relates to the objectives of the MMVAE, MoPoE-VAE, and MVAE without ELBO sub-sampling.

From a computational perspective, a characteristic of mixture-based multimodal VAEs is the sub-sampling of modalities during training, which is a direct consequence of defining the encoder as a mixture distribution over subsets of modalities. The sub-sampling of modalities can be viewed as the extraction of a subset $\boldsymbol{x}_A \in \boldsymbol{x}$, where $A$ indexes one subset of modalities that is drawn from the model-specific set of subsets $\mathcal{S}$. The only member of the family of mixture-based multimodal VAEs that forgoes sub-sampling, defines a trivial mixture over a single subset, the complete set of modalities (Sutter et al., 2021).

## 4 MODALITY SUB-SAMPLING LIMITS THE MULTIMODAL ELBO

### 4.1 AN INTUITION ABOUT THE PROBLEM

Before we delve into the details, let us illustrate how modality sub-sampling affects the likelihood estimation, and hence the multimodal ELBO. Consider the likelihood estimation using the objective $\mathcal{L}_{\mathcal{S}}$:

$$\sum_{A \in \mathcal{S}} \omega_A \, \mathbb{E}_{p(\boldsymbol{x}) p_\theta(\boldsymbol{z} \mid \boldsymbol{x}_A)} [\log q_\phi(\boldsymbol{x} \mid \boldsymbol{z})] \,, \tag{3}$$

where $A$ denotes a subset of modalities and $\omega_A$ the respective mixture weight. Crucially, the stochastic encoder $p_\theta(\boldsymbol{z} \mid \boldsymbol{x}_A)$ encodes a *subset* of modalities. What seems to be a minute detail, can have a profound impact on the likelihood estimation, because the precise estimation of all modalities depends on information from *all* modalities. In trying to reconstruct all modalities from incomplete information, the model can learn an inexact, average prediction; however, it cannot reliably predict modality-specific information, such as the background details in an image given a concise verbal description of its content.

In the following, we formalize the above intuition by showing that, in the presence of modality-specific variation, modality sub-sampling enforces an undesirable upper bound on the multimodal ELBO and therefore prevents a tight approximation of the joint distribution.

### 4.2 A FORMALIZATION OF THE PROBLEM

Theorem 1 states our main theoretical result, which describes a non-trivial limitation of mixture-based multimodal VAEs. Our result shows that the sub-sampling of modalities enforces an undesirable upper bound on the approximation of the joint distribution when there is modality-specific variation in the data. This limitation conflicts with the goal of modeling real-world multimodal data, which typically exhibits a considerable degree of modality-specific variation.

**Theorem 1.** *Each mixture-based multimodal VAE (Definition 3) approximates the expected log-evidence up to an irreducible discrepancy $\Delta(\boldsymbol{X}, \mathcal{S})$ that depends on the model-specific mixture distribution $\mathcal{S}$ as well as on the amount of modality-specific information in $\boldsymbol{X}$.*

*For the maximization of $\mathcal{L}_{\mathcal{S}}(\boldsymbol{x}; \theta, \phi)$ and every value of $\theta$ and $\phi$, the following inequality holds:*

$$\mathbb{E}_{p(\boldsymbol{x})}[\log p(\boldsymbol{x})] \geq \mathcal{L}_{\mathcal{S}}(\boldsymbol{x}; \theta, \phi) + \Delta(\boldsymbol{X}, \mathcal{S}) \tag{4}$$

*where*

$$\Delta(\boldsymbol{X}, \mathcal{S}) = \sum_{A \in \mathcal{S}} \omega_A \, H(\boldsymbol{X}_{\{1,\ldots,M\} \setminus A} \mid \boldsymbol{X}_A) \,. \tag{5}$$

*In particular, the generative discrepancy is always greater than or equal to zero and it is independent of $\theta$ and $\phi$ and thus remains constant during the optimization.*

A proof is provided in Appendix B.5 and it is based on Lemmas 1 and 2. Theorem 1 formalizes the rationale that, in the general case, cross-modal prediction cannot recover information that is specific to the target modalities that are unobserved due to modality sub-sampling. In general, the conditional entropy $H(\boldsymbol{X}_{\{1,\ldots,M\} \setminus A} \mid \boldsymbol{X}_A)$ measures the amount of information in one subset of random vectors $\boldsymbol{X}_{\{1,\ldots,M\} \setminus A}$ that is not shared with another subset $\boldsymbol{X}_A$. In our context, the sub-sampling of modalities yields a discrepancy $\Delta(\boldsymbol{X}, \mathcal{S})$ that is a weighted average of conditional

entropies $H(\boldsymbol{X}_{\{1,...,M\}\setminus A} \mid \boldsymbol{X}_A)$ of the modalities $\boldsymbol{X}_{\{1,...,M\}\setminus A}$ unobserved by the encoder given an observed subset $\boldsymbol{X}_A$. Hence, $\Delta(\boldsymbol{X}, \mathcal{S})$ describes the modality-specific information that cannot be recovered by cross-modal prediction, averaged over all subsets of modalities.

Theorem 1 applies to the MMVAE, MoPoE-VAE, and a special case of the MVAE without ELBO subsampling, since all of these models belong to the class of mixture-based multimodal VAEs. However, $\Delta(\boldsymbol{X}, \mathcal{S})$ can vary significantly between different models, depending on the mixture distribution defined by the respective model and on the amount of modality-specific variation in the data. In the following, we show that without modality sub-sampling $\Delta(\boldsymbol{X}, \mathcal{S})$ vanishes, whereas for the MMVAE and MoPoE-VAE, $\Delta(\boldsymbol{X}, \mathcal{S})$ typically increases with each additional modality. In Section 5, we provide empirical support for each of these theoretical statements.

### 4.3 Implications of Theorem 1

First, we consider the case of no modality sub-sampling, for which it is easy to show that the generative discrepancy vanishes.

**Corollary 1.** *Without modality sub-sampling, $\Delta(\boldsymbol{X}, \mathcal{S}) = 0$.*

A proof is provided in Appendix B.6. The result from Corollary 1 applies to the MVAE without ELBO sub-sampling and suggests that this model should yield a tighter approximation of the joint distribution and hence a better generative quality compared to mixture-based multimodal VAEs that sub-sample modalities. Note that this does not imply that a model without modality sub-sampling is superior to one that uses sub-sampling and that there can be an inductive bias that favors sub-sampling despite the approximation error it incurs. Especially, Corollary 1 does not imply that the variational approximation is tight for the MVAE; for instance, the model can be underparameterized or simply misspecified due to simplifying assumptions, such as the PoE-factorization (Kurle et al., 2019).

Second, we consider how additional modalities might affect the generative discrepancy. Corollary 2 predicts an increased generative discrepancy (and hence, a decline of generative quality) when we increase the number of modalities for the MMVAE and MoPoE-VAE.

**Corollary 2** (informal). *For the MMVAE and MoPoE-VAE, the generative discrepancy increases with each additional modality, if the new modality is sufficiently diverse.*

A proof is provided in Appendix B.7. The notion of *diversity* requires a more formal treatment of the underlying information-theoretic quantities, which we defer to Appendix B.7. Intuitively, a new modality is sufficiently diverse, if it does *not* add too much redundant information with respect to the existing modalities. In special cases when there is a lot of redundant information, $\Delta(\boldsymbol{X}, \mathcal{S})$ can decrease given an additional modality, but it does not vanish in any one of these cases. Only if there is very little modality-specific information in *all* modalities, we have $\Delta(\boldsymbol{X}, \mathcal{S}) \to 0$ for the MMVAE and MoPoE-VAE. This condition requires modalities to be extremely similar, which does not apply to most multimodal datasets, where $\Delta(\boldsymbol{X}, \mathcal{S})$ typically represents a large part of the total variation.

In summary, Theorem 1 formalizes how the family of mixture-based multimodal VAEs is fundamentally limited for the task of approximating the joint distribution, and Corollaries 1 and 2 connect this result to existing models—the MMVAE, MoPoE-VAE, and MVAE without ELBO sub-sampling. We now turn to the experiments, where we present empirical support for the limitations described by Theorem 1 and its Corollaries.

## 5 Experiments

Figure 1 presents the three considered datasets. PolyMNIST (Sutter et al., 2021) is a simple, synthetic dataset with five image modalities that allows us to conduct systematic ablations. Translated-PolyMNIST is a new dataset that adds a small tweak—the downscaling and random translation of digits—to demonstrate the limitations of existing methods when shared information cannot be predicted in expectation across modalities. Finally, Caltech Birds (CUB; Wah et al., 2011; Shi et al., 2019) is used to validate the limitations on a more realistic dataset with two modalities, images and captions. Please note that we use CUB with *real images* and not the simplified version based on precomputed ResNet-features that was used in Shi et al. (2019) and Shi et al. (2021). For a more detailed description of the three considered datasets, please see Appendix C.1.

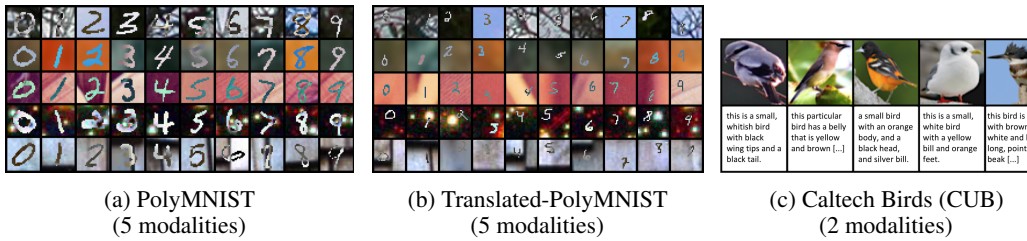

| (a) PolyMNIST | (b) Translated-PolyMNIST | (c) Caltech Birds (CUB) |
| (5 modalities) | (5 modalities) | (2 modalities) |

Figure 1: The three considered datasets. Each subplot shows samples from the respective dataset. The two PolyMNIST datasets are conceptually similar in that the digit label is shared between five synthetic modalities. The Caltech Birds (CUB) dataset provides a more realistic application for which there is no annotation on what is shared between paired images and captions.

In total, more than 400 models were trained, requiring approximately 1.5 GPU years of compute on a single NVIDIA GeForce RTX 2080 Ti GPU. For the experiments in the main text, we use the publicly available code from Sutter et al. (2021) and in Appendix C.3 we also include ablations using the publicly available code from Shi et al. (2019), which implements importance sampling and alternative ELBO objectives. To provide a fair comparison across methods, we use the same architectures and similar capacities for all models. For each unimodal VAE, we make sure to decrease the capacity by reducing the latent dimensionality proportionally with respect to the number of modalities. Additional information on architectures, hyperparameters, and evaluation metrics is provided in Appendix C.

## 5.1 THE GENERATIVE QUALITY GAP

We assume that an increase in the generative discrepancy $\Delta(\boldsymbol{X}, \mathcal{S})$ is associated with a drop of generative quality. However, we want to point out that there can also be an inductive bias that favors modality sub-sampling despite the approximation error that it incurs. In fact, our experiments reveal a fundamental tradeoff between generative quality and generative coherence when shared information can be predicted in expectation across modalities.

We measure generative quality in terms of Fréchet inception distance (FID; Heusel et al., 2017), a standard metric for evaluating the quality of generated images. Lower FID represents better generative quality and the values typically correlate well with human perception (Borji, 2019). In addition, in Appendix C we provide log-likelihood values, as well as qualitative results for all modalities including captions, for which FID cannot be computed.

Figure 2 presents the generative quality across a range of $\beta$ values.[4] To relate different methods, we compare models with the *best* FID respectively, because different methods can reach their optima at different $\beta$ values. As described by Theorem 1, mixture-based multimodal VAEs that sub-sample modalities (MMVAE and MoPoE-VAE) exhibit a pronounced generative quality gap compared to unimodal VAEs. When we compare the best models, we observe a gap of more than 60 points on both PolyMNIST and Translated-PolyMNIST, and about 30 points on CUB images. Qualitative results (Figure 9 in Appendix C.3) confirm that this gap is clearly visible in the generated samples and that it applies not only to image modalities, but also to captions. In contrast, the MVAE (without ELBO sub-sampling) reaches the generative quality of unimodal VAEs, which is in line with our theoretical result from Corollary 1. For completeness, in Appendix C.3, we also report joint log-likelihoods, latent classification performance, as well as additional FIDs for all modalities.

Figure 3 examines how the generative quality is affected when we vary the number of modalities. Notably, for the MMVAE and MoPoE-VAE, the generative quality deteriorates almost continuously with the number of modalities, which is in line with our theoretical result from Corollary 2. Interestingly, for the MVAE, the generative quality on Translated-PolyMNIST also decreases slightly, but the change is comparatively small. Figure 11 in Appendix C.3, shows a similar trend even when we control for modality-specific differences by generating PolyMNIST using the *same* background image for all modalities.

---

[4]The regularization coefficient $\beta$ weights the KL-divergence term of the multimodal ELBO (Definitions 1 and 3) and it is arguably the most impactful hyperparameter in VAEs (e.g., see Higgins et al., 2017).

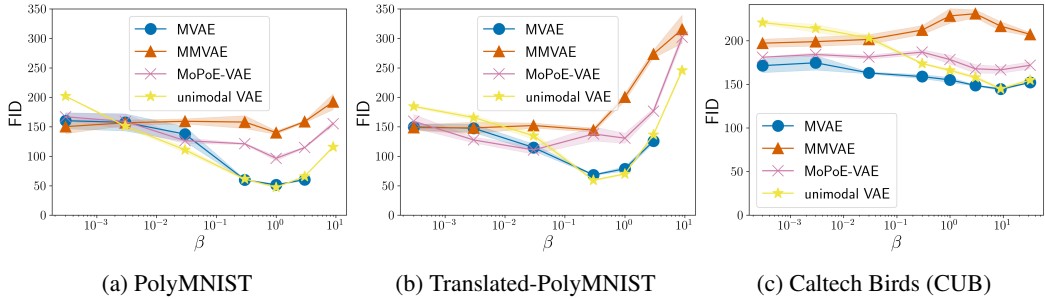

(a) PolyMNIST                (b) Translated-PolyMNIST          (c) Caltech Birds (CUB)

Figure 2: Generative quality for one output modality over a range of $\beta$ values. Points denote the FID averaged over three seeds and bands show one standard deviation respectively. Due to numerical instabilities, the MVAE could not be trained with larger $\beta$ values.

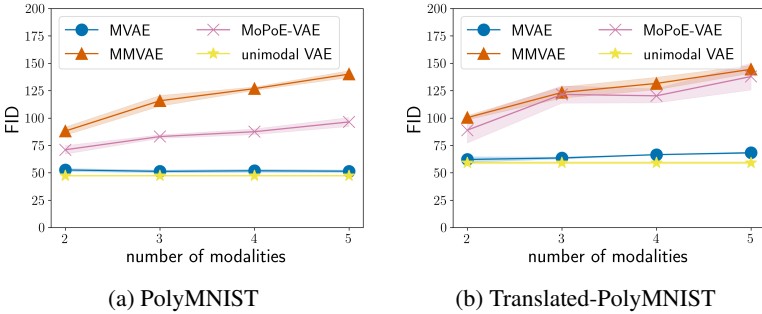

(a) PolyMNIST                        (b) Translated-PolyMNIST

Figure 3: Generative quality as a function of the number of modalities. The results show the FID of the same modality and therefore all values on the same scale. All models are trained with $\beta = 1$ on PolyMNIST and $\beta = 0.3$ on Translated-PolyMNIST. The results are averaged over three seeds and the bands show one standard deviation respectively. For the unimodal VAE, which uses only a single modality, the average and standard deviation are plotted as a constant.

In summary, the results from Figure 2 and Figure 3 provide empirical support for the existence of a generative quality gap between unimodal and mixture-based multimodal VAEs that sub-sample modalities. The results verify that the approximation of the joint distribution improves for models without sub-sampling, which manifests in better generative quality. In contrast, the gap increases disproportionally with each additional modality for both the MMVAE and MoPoE-VAE. Hence, the presented results support all of the theoretical statements from Sections 4.2 and 4.3.

## 5.2    LACK OF GENERATIVE COHERENCE ON MORE COMPLEX DATA

Apart from generative quality, another desired criterion (Shi et al., 2019; Sutter et al., 2020) for an effective multimodal generative model is *generative coherence*, which measures a model's ability to generate semantically related samples across modalities. To be consistent with Sutter et al. (2021), we compute the leave-one-out coherence (see Appendix C.2), which means that the input to each model consists of all modalities except the one that is being conditionally generated. On CUB, we resort to a qualitative evaluation of coherence, because there is no ground truth annotation of shared factors and the proxies used in Shi et al. (2019) and Shi et al. (2021) do not yield meaningful estimates when applied to the conditionally generated images from models that were trained on *real* images.[5]

In terms of generative coherence, Figure 4 reveals that the positive results from previous work do not translate to more complex datasets. As a baseline, for PolyMNIST (Figure 4a) we replicate the coherence results from Sutter et al. (2021) for a range of $\beta$ values. Consistent with previous work (Shi et al., 2019; 2021; Sutter et al., 2020; 2021), we find that the MMVAE and MoPoE-VAE exhibit

---

[5]Please note that previous work (Shi et al., 2019; 2021) used a simplified version of the CUB dataset, where images were replaced by precomputed ResNet-features.

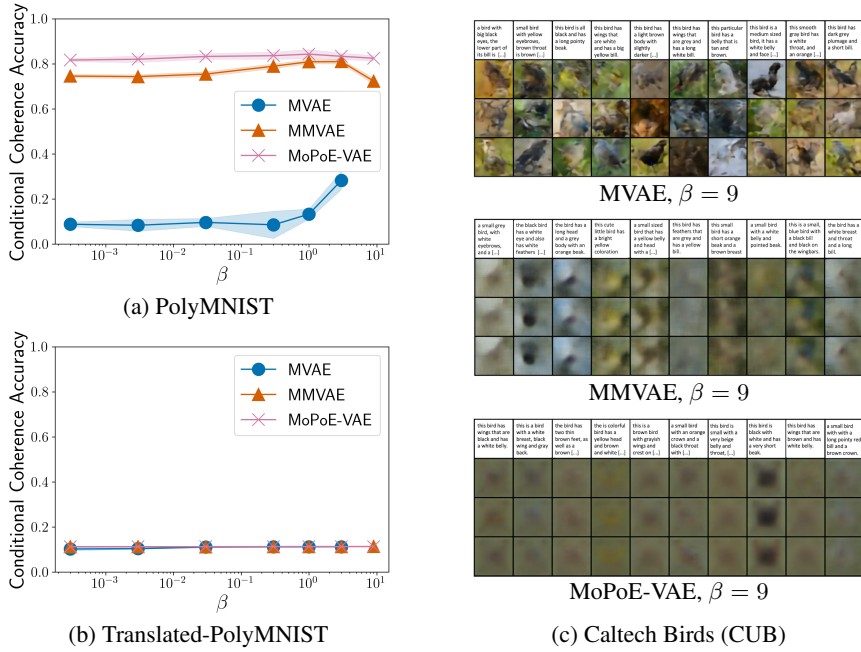

(a) PolyMNIST

(b) Translated-PolyMNIST

MVAE, $\beta = 9$

MMVAE, $\beta = 9$

MoPoE-VAE, $\beta = 9$

(c) Caltech Birds (CUB)

Figure 4: Generative coherence for the conditional generation across modalities. For PolyMNIST (Figures 4a and 4b), we plot the average leave-one-out coherence. Due to numerical instabilities, the MVAE could not be trained with larger $\beta$ values. For CUB (Figure 4c), we show qualitative results for the conditional generation of images given captions. Best viewed zoomed and in color.

superior coherence compared to the MVAE. Though, it was not apparent from previous work that MVAE's coherence can improve significantly with increasing $\beta$ values, which can be of independent interest for future work. On Translated-PolyMNIST (Figure 4b), the stark decline of all models makes it evident that coherence cannot be guaranteed when shared information cannot be predicted in expectation across modalities. Our qualitative results (Figure 10 in Appendix C.3) confirm that not a single multimodal VAE is able to conditionally generate coherent examples and, for the most part, not any digits at all. To verify that the lack of coherence is not an artifact of our implementation, we have checked that the encoders and decoders have sufficient capacity such that digits show up in most self-reconstructions. On CUB (Figure 4c), for which coherence cannot be computed, the qualitative results for conditional generation verify that none of the existing approaches generates images that are both of sufficiently high quality and coherent with respect to the given caption. Overall, the negative results on Translated-PolyMNIST and CUB showcase the limitations of existing approaches when applied to more complex datasets than those used in previous benchmarks.

## 6 DISCUSSION

**Implications and scope** Our experiments lend empirical support to the proposed theoretical limitations of mixture-based multimodal VAEs. On both synthetic and real data, our results showcase the generative limitations of multimodal VAEs that sub-sample modalities. However, our results also reveal that none of the existing approaches (including those without sub-sampling) fulfill all desired criteria (Shi et al., 2019; Sutter et al., 2020) of an effective multimodal generative model. More broadly, our results showcase the limitations of existing VAE-based approaches for modeling weakly-supervised data in the presence of modality-specific information, and in particular when shared information cannot be predicted in expectation across modalities. The Translated-PolyMNIST dataset demonstrates this problem in a simple setting, while the results on CUB confirm that similar issues can be expected on more realistic datasets. For future work, it would be interesting to generate simulated data where the discrepancy $\Delta(\boldsymbol{X}, \mathcal{S})$ can be measured exactly and where it is gradually increased by an adaptation of the dataset in a way that increases only the modality-specific variation. Furthermore, it is worth noting that Theorem 1 applies to all multimodal VAEs that optimize Equa-

tion (2), which is a lower bound on the multimodal ELBO for models that sub-sample modalities. Our theory predicts the same discrepancy for models that optimize a tighter bound (e.g., via Equation (28)), because the discrepancy $\Delta(\boldsymbol{X}, \mathcal{S})$ derives from the likelihood term, which is equal for Equations (2) and (28). In Appendix C.3 we verify that the discrepancy can also be observed for the MMVAE with the original implementation from Shi et al. (2019) that uses a tighter bound. Further analysis of the different bounds can be an interesting direction for future work.

**Model selection and generalization**  Our results raise fundamental questions regarding model selection and generalization, as generative quality and generative coherence do not necessarily go hand in hand. In particular, our experiments demonstrate that FIDs and log-likelihoods do not reflect the problem of lacking coherence and without access to ground truth labels (on what is shared between modalities) coherence metrics cannot be computed. As a consequence, it can be difficult to perform model selection on more realistic multimodal datasets, especially for less interpretable types of modalities, such as DNA sequences. Hence, for future work it would be interesting to design alternative metrics for generative coherence that can be applied when shared information is not annotated. For the related topic of generalization, it can be illuminating to consider what would happen, if one could arbitrarily "scale things up". In the limit of infinite i.i.d. data, perfect generative coherence could be achieved by a model that memorizes the pairwise relations between training examples from different modalities. However, would this yield a model that generalizes out of distribution (e.g., under distribution shift)? We believe that for future work it would be worthwhile to consider out-of-distribution generalization performance (e.g., Montero et al., 2021) in addition to generative quality and coherence.

**Limitations**  In general, the limitations and tradeoffs presented in this work apply to a large family of multimodal VAEs, but not necessarily to other types of generative models, such as generative adversarial networks (Goodfellow et al., 2014). Where current VAEs are limited by the reconstruction of modality-specific information, other types of generative models might offer less restrictive objectives. Similar to previous work, we have only considered models with simple priors, such as Gauss and Laplace distributions with independent dimensions. Further, we have not considered models with modality-specific latent spaces, which seem to yield better empirical results (Hsu and Glass, 2018; Sutter et al., 2020; Daunhawer et al., 2020), but currently lack theoretical grounding. Modality-specific latent spaces offer a potential solution to the problem of cross-modal prediction by providing modality-specific context from the target modalities to each decoder. However, more work is required to establish *guarantees* for the identifiability and disentanglement of shared and modality-specific factors, which might only be possible for VAEs under relatively strong assumptions (Locatello et al., 2019; 2020; Gresele et al., 2019; von Kügelgen et al., 2021).

## 7  CONCLUSION

In this work, we have identified, formalized, and demonstrated several limitations of multimodal VAEs. Across different datasets, this work revealed a significant gap in generative quality between unimodal and mixture-based multimodal VAEs. We showed that this apparent paradox can be explained by the sub-sampling of modalities, which enforces an undesirable upper bound on the multimodal ELBO and therefore limits the generative quality of the respective models. While the sub-sampling of modalities allows these models to learn the inference networks for different subsets of modalities efficiently, there is a notable tradeoff in terms of generative quality. Finally, we studied two failure cases—Translated-PolyMNIST and CUB—that demonstrate the limitations of multimodal VAEs when applied to more complex datasets than those used in previous benchmarks.

For future work, we believe that it is crucial to be aware of the limitations of existing methods as a first step towards developing new methods that achieve more than incremental improvements for multimodal learning. We conjecture that there are at least two potential strategies to circumvent the theoretical limitations of multimodal VAEs. First, the sub-sampling of modalities can be combined with modality-specific context from the target modalities. Second, cross-modal reconstruction terms can be replaced with less restrictive objectives that do not require an exact prediction of modality-specific information. Finally, we urge future research to design more challenging benchmarks and to compare multimodal generative models in terms of both generative quality and coherence across a range of hyperparameter values, to present the tradeoff between these metrics more transparently.

ACKNOWLEDGEMENTS

ID and KC were supported by the SNSF grant `#200021_188466`. Special thanks to Alexander Marx, Nicolò Ruggeri, Maxim Samarin, Yuge Shi, and Mario Wieser for helpful discussions and/or feedback on the manuscript.

REPRODUCIBILITY STATEMENT

For all theoretical statements, we provide detailed derivations and state the necessary assumptions. For our main theoretical results, we present empirical support on both synthetic and real data. To ensure empirical reproducibility, the results of each experiment and every ablation were averaged over multiple seeds and are reported with standard deviations. All of the used datasets are either public or can be generated from publicly available resources using the code that we provide in the supplementary material. Information about implementation details, hyperparameter settings, and evaluation metrics are included in Appendix C.

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

# A DEFINITIONS

Let $\mathcal{X}$, $\mathcal{Y}$, and $\mathcal{Z}$ denote the support sets of three discrete random vectors $\boldsymbol{X}$, $\boldsymbol{Y}$, and $\boldsymbol{Z}$ respectively. Let $p_{\boldsymbol{X}}(\boldsymbol{x})$, $p_{\boldsymbol{Y}}(\boldsymbol{y})$, and $p_{\boldsymbol{Z}}(\boldsymbol{z})$ denote the respective marginal distributions and note that we will leave out the subscripts (e.g., $p(\boldsymbol{x})$ instead of $p_{\mathcal{X}}(\boldsymbol{x})$) when it is clear from context which distribution we are referring to. Analogously, we write shorthand $p(\boldsymbol{y} \mid \boldsymbol{x})$ for the conditional distribution of $\boldsymbol{Y}$ given $\boldsymbol{X}$ and $p(\boldsymbol{x}, \boldsymbol{y})$ for the joint distribution of $\boldsymbol{X}$ and $\boldsymbol{Y}$.

The entropy of $\boldsymbol{X}$ is defined as

$$H(\boldsymbol{X}) = - \sum_{\boldsymbol{x} \in \mathcal{X}} p(\boldsymbol{x}) \log p(\boldsymbol{x}) \ . \tag{6}$$

The conditional entropy of $\boldsymbol{X}$ given $\boldsymbol{Y}$ is defined as

$$H(\boldsymbol{X} \mid \boldsymbol{Y}) = - \sum_{\boldsymbol{x} \in \mathcal{X}, \boldsymbol{y} \in \mathcal{Y}} p(\boldsymbol{x}, \boldsymbol{y}) \log p(\boldsymbol{x} \mid \boldsymbol{y}) \ . \tag{7}$$

The joint entropy of $\boldsymbol{X}$ and $\boldsymbol{Y}$ is defined as

$$H(\boldsymbol{X}, \boldsymbol{Y}) = - \sum_{\boldsymbol{x} \in \mathcal{X}, \boldsymbol{y} \in \mathcal{Y}} p(\boldsymbol{x}, \boldsymbol{y}) \log p(\boldsymbol{x}, \boldsymbol{y}) \ . \tag{8}$$

The Kullback-Leibler divergence of the discrete probability distribution $P$ from the discrete probability distribution $Q$ is defined as

$$D_{\mathrm{KL}}(P \,\|\, Q) = \sum_{\boldsymbol{x} \in \mathcal{X}} P(\boldsymbol{x}) \log \frac{P(\boldsymbol{x})}{Q(\boldsymbol{x})} \tag{9}$$

assuming that $P$ and $Q$ are defined on the same support set $\mathcal{X}$.

The cross-entropy of the discrete probability distribution $Q$ from the discrete probability distribution $P$ is defined as

$$CE(P, Q) = - \sum_{\boldsymbol{x} \in \mathcal{X}} P(\boldsymbol{x}) \log Q(\boldsymbol{x}) \tag{10}$$

assuming that $P$ and $Q$ are defined on the same support set $\mathcal{X}$.

The mutual information of $\boldsymbol{X}$ and $\boldsymbol{Y}$ is defined as

$$I(\boldsymbol{X}; \boldsymbol{Y}) = D_{\mathrm{KL}}(p(\boldsymbol{x}, \boldsymbol{y}) \,\|\, p(\boldsymbol{x}) p(\boldsymbol{y})) \ . \tag{11}$$

The conditional mutual information of $\boldsymbol{X}$ and $\boldsymbol{Y}$ given $\boldsymbol{Z}$ is defined as

$$I(\boldsymbol{X}; \boldsymbol{Y} \mid \boldsymbol{Z}) = \sum_{\boldsymbol{z} \in \mathcal{Z}} p(\boldsymbol{z}) D_{\mathrm{KL}}(p(\boldsymbol{x}, \boldsymbol{y} \mid \boldsymbol{z}) \,\|\, p(\boldsymbol{x} \mid \boldsymbol{z}) p(\boldsymbol{y} \mid \boldsymbol{z})) \ . \tag{12}$$

Recall that we assume discrete random vectors (e.g., pixel values) and therefore can assume non-negative entropy, conditional entropy and conditional mutual information terms (Cover and Thomas, 2012). For continuous random variables, all of the above sums can be replaced with integrals. The only information-theoretic quantities for which in this work we use continuous random vectors are the KL-divergence and mutual information, both of which are always non-negative.

## B PROOFS

### B.1 INFORMATION-THEORETIC DERIVATION OF THE MULTIMODAL ELBO

Proposition 1 relates the multimodal ELBO (Definition 1) to the expected log-evidence, the quantity that is being approximated by all likelihood-based generative models including VAEs. The derivation is based on a straightforward extension of the variational information bottleneck (VIB; Alemi et al., 2017). We include the result mainly for the purpose of illustration—to clarify the notation, as well as the relation between the multimodal ELBO and the underlying information-theoretic quantities of interest: the entropy, conditional entropy, and mutual information.

**Notation** Readers who are familiar with latent variable models, but may be less familiar with the information-theoretic perspective on VAEs, please keep in mind the following notational differences. In contrast to the latent variable model perspective, which defines a variational posterior (typically denoted by the letter $q$) and a stochastic decoder (typically denoted by the letter $p$), the VIB defines a stochastic encoder $p_\theta(z \mid x)$ and variational decoder $q_\phi(x \mid z)$. Moreover, the VIB makes no assumptions about the true posterior. Also note that latent variable models tend to write the ELBO with respect to the log-evidence $\log p(x)$, but information-theoretic approaches write the ELBO with respect to the *expected* log-evidence $\mathbb{E}_{p(x)}[\log p(x)]$; though, it is still assumed that the estimation of the ELBO is based on a finite sample from $p(x)$.

**Proposition 1.** *The multimodal ELBO forms a variational lower bound on the expected log-evidence:*

$$\mathbb{E}_{p(x)}[\log p(x)] \geq \mathcal{L}(x; \theta, \phi) . \tag{13}$$

*Proof.* First, notice that the expected log-evidence is equal to the negative entropy $-H(X) = \mathbb{E}_{p(x)}[\log p(x)]$. Given any random variable $Z$, the entropy can be decomposed into conditional entropy and mutual information terms: $H(X) = H(X \mid Z) + I(X; Z)$.

The expected log-evidence relates to the multimodal ELBO as follows:

$$\mathbb{E}_{p(x)}[\log p(x)] = -H(X \mid Z) - I(X; Z) \tag{14}$$

$$\geq \mathbb{E}_{p(x)p_\theta(z \mid x)}[\log q_\phi(x \mid z)] - \mathbb{E}_{p(x)}[D_{\mathrm{KL}}(p_\theta(z \mid x) \,||\, q(z))] \tag{15}$$

$$= \mathcal{L}(x; \theta, \phi) \tag{16}$$

where the inequality follows from the variational approximations of the respective terms. As in Alemi et al. (2017), we can use the following variational bounds.

For the conditional entropy, we have

$$-H(X \mid Z) = \mathbb{E}_{p(x)p_\theta(z \mid x)} [\log p(x \mid z)] \tag{17}$$

$$= \mathbb{E}_{p(x)p_\theta(z \mid x)} [\log q_\phi(x \mid z)] + \mathbb{E}_{p(z)} [D_{\mathrm{KL}}(p(x \mid z) \,||\, q_\phi(x \mid z))] \tag{18}$$

$$\geq \mathbb{E}_{p(x)p_\theta(z \mid x)} [\log q_\phi(x \mid z)] \tag{19}$$

where $q_\phi(x \mid z)$ is a variational decoder that is parameterized by $\phi$.

For the mutual information, we have

$$-I(X; Z) = -\mathbb{E}_{p(x)} [D_{\mathrm{KL}}(p_\theta(z \mid x) \,||\, p(z))] \tag{20}$$

$$= -\mathbb{E}_{p(x)} [D_{\mathrm{KL}}(p_\theta(z \mid x) \,||\, q(z))] + D_{\mathrm{KL}}(p(z) \,||\, q(z)) \tag{21}$$

$$\geq -\mathbb{E}_{p(x)} [D_{\mathrm{KL}}(p_\theta(z \mid x) \,||\, q(z))] \tag{22}$$

where $q(z)$ is a prior.

Hence, the multimodal ELBO forms a variational lower bound on the expected log-evidence:

$$\mathbb{E}_{p(x)}[\log p(x)] = \mathcal{L}(x; \theta, \phi) + \Delta_{\mathrm{VA}}(x, \phi) \tag{23}$$

$$\geq \mathcal{L}(x; \theta, \phi) \tag{24}$$

where

$$\Delta_{\mathrm{VA}}(x, \phi) = \mathbb{E}_{p(z)} [D_{\mathrm{KL}}(p(x \mid z) \,||\, q_\phi(x \mid z))] + D_{\mathrm{KL}}(p(z) \,||\, q(z)) \tag{25}$$

denotes the (non-negative) variational approximation gap.

$\square$

### B.2 RELATION BETWEEN THE DIFFERENT OBJECTIVES

Proposition 2 relates the multimodal ELBO $\mathcal{L}$ from Definition 1 to the objective $\mathcal{L}_{\mathcal{S}}$, which is a general formulation of the objective maximized by all mixture-based multimodal VAEs. Compared to previous mixture-based formulations (Shi et al., 2019; Sutter et al., 2020), our formulation is more general in that it allows for arbitrary subsets with non-uniform mixture coefficients. Further, the derivation *quantifies* the approximation gap between $\mathcal{L}$ and $\mathcal{L}_{\mathcal{S}}$, where the latter corresponds to the objectives that are actually being optimized in the implementations of the MMVAE, MoPoE-VAE, and MVAE without sub-sampling.

**Proposition 2.** *For every stochastic encoder $p_\theta^{\mathcal{S}}(\boldsymbol{z} \mid \boldsymbol{x})$ that is consistent with Definition 2, the following inequality holds:*

$$\mathcal{L}(\boldsymbol{x}; \theta, \phi) \geq \mathcal{L}_{\mathcal{S}}(\boldsymbol{x}; \theta, \phi) . \tag{26}$$

*Proof.* Recall the multimodal ELBO from Definition 1:

$$\mathcal{L}(\boldsymbol{x}; \theta, \phi) = \mathbb{E}_{p(\boldsymbol{x})p_\theta(\boldsymbol{z} \mid \boldsymbol{x})}[\log q_\phi(\boldsymbol{x} \mid \boldsymbol{z})] - \mathbb{E}_{p(\boldsymbol{x})}[D_{\mathrm{KL}}(p_\theta(\boldsymbol{z} \mid \boldsymbol{x}) \,\|\, q(\boldsymbol{z}))] . \tag{27}$$

For the encoder $p_\theta(\boldsymbol{z} \mid \boldsymbol{x})$, plug in the mixture-based encoder $p_\theta^{\mathcal{S}}(\boldsymbol{z} \mid \boldsymbol{x}) = \sum_{A \in \mathcal{S}} \omega_A\, p_\theta(\boldsymbol{z} \mid \boldsymbol{x}_A)$ from Definition 2 and re-write as follows:

$$\mathbb{E}_{p(\boldsymbol{x})p_\theta^{\mathcal{S}}(\boldsymbol{z} \mid \boldsymbol{x})}[\log q_\phi(\boldsymbol{x} \mid \boldsymbol{z})] - \mathbb{E}_{p(\boldsymbol{x})}[D_{\mathrm{KL}}(p_\theta^{\mathcal{S}}(\boldsymbol{z} \mid \boldsymbol{x}) \,\|\, q(\boldsymbol{z}))] \tag{28}$$

$$= \mathbb{E}_{p(\boldsymbol{x}) \sum_{A \in \mathcal{S}} \omega_A\, p_\theta(\boldsymbol{z} \mid \boldsymbol{x}_A)}[\log q_\phi(\boldsymbol{x} \mid \boldsymbol{z})] - \tag{29}$$
$$\mathbb{E}_{p(\boldsymbol{x}) \sum_{A \in \mathcal{S}} \omega_A\, p_\theta(\boldsymbol{z} \mid \boldsymbol{x}_A)}[\log p_\theta^{\mathcal{S}}(\boldsymbol{z} \mid \boldsymbol{x}) - \log q(\boldsymbol{z})]$$

$$= \sum_{A \in \mathcal{S}} \omega_A \big\{ \mathbb{E}_{p(\boldsymbol{x})p_\theta(\boldsymbol{z} \mid \boldsymbol{x}_A)}[\log q_\phi(\boldsymbol{x} \mid \boldsymbol{z})] - \mathbb{E}_{p(\boldsymbol{x})p_\theta(\boldsymbol{z} \mid \boldsymbol{x}_A)}[\log p_\theta^{\mathcal{S}}(\boldsymbol{z} \mid \boldsymbol{x})] + \tag{30}$$
$$\mathbb{E}_{p(\boldsymbol{x})p_\theta(\boldsymbol{z} \mid \boldsymbol{x}_A)}[\log q(\boldsymbol{z})] \big\}$$

$$= \sum_{A \in \mathcal{S}} \omega_A \big\{ \mathbb{E}_{p(\boldsymbol{x})p_\theta(\boldsymbol{z} \mid \boldsymbol{x}_A)}[\log q_\phi(\boldsymbol{x} \mid \boldsymbol{z})] + \mathbb{E}_{p(\boldsymbol{x})}[CE(p_\theta(\boldsymbol{z} \mid \boldsymbol{x}_A), p_\theta^{\mathcal{S}}(\boldsymbol{z} \mid \boldsymbol{x}))] - \tag{31}$$
$$\mathbb{E}_{p(\boldsymbol{x})}[CE(p_\theta(\boldsymbol{z} \mid \boldsymbol{x}_A), q(\boldsymbol{z}))] \big\}$$

$$= \sum_{A \in \mathcal{S}} \omega_A \big\{ \mathbb{E}_{p(\boldsymbol{x})p_\theta(\boldsymbol{z} \mid \boldsymbol{x}_A)}[\log q_\phi(\boldsymbol{x} \mid \boldsymbol{z})] + \mathbb{E}_{p(\boldsymbol{x})}[D_{\mathrm{KL}}(p_\theta(\boldsymbol{z} \mid \boldsymbol{x}_A) \,\|\, p_\theta^{\mathcal{S}}(\boldsymbol{z} \mid \boldsymbol{x}))] - \tag{32}$$
$$\mathbb{E}_{p(\boldsymbol{x})}[D_{\mathrm{KL}}(p_\theta(\boldsymbol{z} \mid \boldsymbol{x}_A) \,\|\, q(\boldsymbol{z}))] \big\}$$

$$\geq \sum_{A \in \mathcal{S}} \omega_A \big\{ \mathbb{E}_{p(\boldsymbol{x})p_\theta(\boldsymbol{z} \mid \boldsymbol{x}_A)}[\log q_\phi(\boldsymbol{x} \mid \boldsymbol{z})] - \mathbb{E}_{p(\boldsymbol{x})}[D_{\mathrm{KL}}(p_\theta(\boldsymbol{z} \mid \boldsymbol{x}_A) \,\|\, q(\boldsymbol{z}))] \big\} \tag{33}$$

$$= \mathcal{L}_{\mathcal{S}}(\boldsymbol{x}; \theta, \phi) \tag{34}$$

In Equation (31), $CE(p, q)$ denotes the cross-entropy between distributions $p$ and $q$. For Equation (32), decompose both cross-entropy terms using $CE(p, q) = H(p) + D_{\mathrm{KL}}(p \,\|\, q)$ and notice that the respective entropy terms cancel out. The inequality (Equation (33)) follows from the non-negativity of the KL-divergence. This concludes the proof that $\mathcal{L}_{\mathcal{S}}(\boldsymbol{x}; \theta, \phi)$ forms a lower bound on $\mathcal{L}(\boldsymbol{x}; \theta, \phi)$.

$\square$

**Objectives of individual models**  Sutter et al. (2021) already showed that Equation (28) subsumes the objectives of the MMVAE, MoPoE-VAE, and MVAE without ELBO sub-sampling. However, in their actual implementation, all of these methods take the sum out of the KL-divergence term (e.g., see Shi et al., 2019, Equation 3), which corresponds to the objective $\mathcal{L}_{\mathcal{S}}$. To see how $\mathcal{L}_{\mathcal{S}}$ recovers the objectives of the individual models, simply plug in the model-specific definition of $\mathcal{S}$ into Equation (33) and use uniform mixture coefficients $\omega_A = 1/|\mathcal{S}|$ for all subsets. For the MVAE without ELBO sub-sampling, $\mathcal{S}$ is comprised of only one subset, the complete set of modalities $\{\boldsymbol{x}_1, \ldots, \boldsymbol{x}_M\}$. For the MMVAE, $\mathcal{S}$ is comprised of the set of unimodal subsets $\{\{\boldsymbol{x}_1\}, \ldots, \{\boldsymbol{x}_M\}\}$. For the MoPoE-VAE, $\mathcal{S}$ is comprised of the powerset $\mathcal{P}(M) \setminus \{\emptyset\}$. Further implementation details, such as importance sampling and ELBO sub-sampling, are discussed in Appendix C.3.

### B.3 OBJECTIVE $\mathcal{L}_{\mathcal{S}}$ IS A SPECIAL CASE OF THE VIB

**Lemma 1.** *$\mathcal{L}_{\mathcal{S}}(\boldsymbol{x}; \theta, \phi)$ is a special case of the variational information bottleneck (VIB) objective*

$$\min_{\psi} \sum_{A \in \mathcal{S}} \omega_A \left\{ H_\psi(\boldsymbol{X} \mid Z_A) + I_\psi(\boldsymbol{X}_A; Z_A) \right\} , \tag{35}$$

*where the encoding $Z_A = f_\psi(\boldsymbol{X}_A)$ is a function of a subset $\boldsymbol{X}_A$, the terms $H_\psi(\boldsymbol{X} \mid Z_A)$ and $I_\psi(\boldsymbol{X}_A; Z_A)$ denote variational upper bounds of $H(\boldsymbol{X} \mid Z_A)$ and $I(\boldsymbol{X}_A; Z_A)$ respectively, and $\psi$ summarizes the parameters of these variational estimators.*

*Proof.* We start from $\mathcal{L}_{\mathcal{S}}$, the objective optimized by all mixture-based multimodal VAEs. Recall from Definition 3:

$$\mathcal{L}_{\mathcal{S}}(\boldsymbol{x}; \theta, \phi) = \sum_{A \in \mathcal{S}} \omega_A \bigg\{ \underbrace{\mathbb{E}_{p(\boldsymbol{x})p_\theta(\boldsymbol{z} \mid \boldsymbol{x}_A)}[\log q_\phi(\boldsymbol{x} \mid \boldsymbol{z})]}_{(i)} - \underbrace{\mathbb{E}_{p(\boldsymbol{x})}\left[D_{\mathrm{KL}}\left(p_\theta(\boldsymbol{z} \mid \boldsymbol{x}_A) \,\|\, q(\boldsymbol{z})\right)\right]}_{(ii)} \bigg\} . \tag{36}$$

Each term within the sum is comprised of two terms: $(i)$ the log-likelihood estimation based on a variational decoder $q_\phi(\boldsymbol{x} \mid \boldsymbol{z})$; $(ii)$ the regularization of the stochastic encoder $p_\theta(\boldsymbol{z} \mid \boldsymbol{x}_A)$ with respect to a variational prior $q(\boldsymbol{z})$. The sampled encoding $\boldsymbol{z} \sim p_\theta(\boldsymbol{z} \mid \boldsymbol{x}_A)$ can be viewed as the output of a function $Z_A = f_\theta(\boldsymbol{X}_A)$ of a subset of modalities.

To see the relation to the underlying information terms $H(\boldsymbol{X} \mid Z_A)$ and $I(\boldsymbol{X}_A; Z_A)$, we undo the variational approximation for $(i)$ and $(ii)$ by re-introducing the unobserved ground truth decoder $p(\boldsymbol{x} \mid \boldsymbol{z})$ and the ground truth prior $p(\boldsymbol{z})$.

For $(i)$, we have

$$\mathbb{E}_{p(\boldsymbol{x})p_\theta(\boldsymbol{z} \mid \boldsymbol{x}_A)}\left[\log q_\phi(\boldsymbol{x} \mid \boldsymbol{z})\right] \leq \mathbb{E}_{p(\boldsymbol{x})p_\theta(\boldsymbol{z} \mid \boldsymbol{x}_A)}\left[\log q_\phi(\boldsymbol{x} \mid \boldsymbol{z})\right] + \tag{37}$$
$$\mathbb{E}_{p(\boldsymbol{z})}\left[D_{\mathrm{KL}}(p(\boldsymbol{x} \mid \boldsymbol{z}) \,\|\, q_\phi(\boldsymbol{x} \mid \boldsymbol{z}))\right]$$
$$= \mathbb{E}_{p(\boldsymbol{x})p_\theta(\boldsymbol{z} \mid \boldsymbol{x}_A)}\left[\log p(\boldsymbol{x} \mid \boldsymbol{z})\right] \tag{38}$$
$$= -H(\boldsymbol{X} \mid Z_A) \tag{39}$$

For $(ii)$, we have

$$\mathbb{E}_{p(\boldsymbol{x})}\left[D_{\mathrm{KL}}(p_\theta(\boldsymbol{z} \mid \boldsymbol{x}_A) \,\|\, q(\boldsymbol{z}))\right] \geq \mathbb{E}_{p(\boldsymbol{x})}\left[D_{\mathrm{KL}}(p_\theta(\boldsymbol{z} \mid \boldsymbol{x}_A) \,\|\, q(\boldsymbol{z}))\right] - D_{\mathrm{KL}}(p(\boldsymbol{z}) \,\|\, q(\boldsymbol{z})) \tag{40}$$
$$= \mathbb{E}_{p(\boldsymbol{x})}\left[D_{\mathrm{KL}}(p_\theta(\boldsymbol{z} \mid \boldsymbol{x}_A) \,\|\, p(\boldsymbol{z}))\right] \tag{41}$$
$$= I(\boldsymbol{X}_A; Z_A) \tag{42}$$

Since $\mathcal{L}_{\mathcal{S}}(\boldsymbol{x}; \theta, \phi)$ is being maximized, $(i)$ is being maximized, while $(ii)$ is being minimized. The maximization of $(i)$ is equal to the minimization of a variational upper bound on $H(\boldsymbol{X} \mid Z_A)$. Similarly, the minimization of $(ii)$ is equal to the minimization of a variational upper bound on $I(\boldsymbol{X}_A; Z_A)$. Hence, we have established that $\mathcal{L}_{\mathcal{S}}(\boldsymbol{x}; \theta, \phi)$ is a special case of the more general VIB objective (Equation (35)) where the information terms are estimated with a mixture-based multimodal VAE that is parameterized by $\psi = \{\theta, \phi\}$.

$\square$

### B.4 DECOMPOSITION OF THE CONDITIONAL ENTROPY FOR SUBSETS OF MODALITIES

**Lemma 2.** *Let $\boldsymbol{X}_A \subseteq \boldsymbol{X}$ be some subset of modalites. If $Z_A = f(\boldsymbol{X}_A)$, where $f$ is some function of the subset $\boldsymbol{X}_A$, then the following equality holds:*

$$H(\boldsymbol{X} \mid Z_A) = H(\boldsymbol{X}_{\{1,\dots,M\} \setminus A} \mid \boldsymbol{X}_A) + H(\boldsymbol{X}_A \mid Z_A) . \tag{43}$$

*Proof.* When $Z_A$ is a function of a subset $\boldsymbol{X}_A \subseteq \boldsymbol{X}$, we have the Markov chain $Z_A \leftarrow \boldsymbol{X}_A - \boldsymbol{X}_{\{1,\dots,M\} \setminus A}$, since $Z_A$ is a function of the (observed) subset of modalities and depends on the remaining (unobserved) modalities only through $\boldsymbol{X}_A$.

We can re-write $H(\boldsymbol{X} \mid Z_A)$ as follows:

$$H(\boldsymbol{X} \mid Z_A) = H(\boldsymbol{X} \mid Z_A, \boldsymbol{X}_A) + I(\boldsymbol{X}; \boldsymbol{X}_A \mid Z_A) \tag{44}$$

$$= H(\boldsymbol{X} \mid \boldsymbol{X}_A) + I(\boldsymbol{X}; \boldsymbol{X}_A \mid Z_A) \tag{45}$$

$$= H(\boldsymbol{X}_{\{1,\ldots,M\}\backslash A} \mid \boldsymbol{X}_A) + I(\boldsymbol{X}; \boldsymbol{X}_A \mid Z_A) \tag{46}$$

$$= H(\boldsymbol{X}_{\{1,\ldots,M\}\backslash A} \mid \boldsymbol{X}_A) + H(\boldsymbol{X}_A \mid Z_A) \tag{47}$$

Equation (44) applies the definition of the conditional mutual information. Equation (45) is based on the conditional independence $\boldsymbol{X} \perp\!\!\!\perp Z_A \mid \boldsymbol{X}_A$ implied by the Markov chain. Equation (46) removes the "known" information that we condition on. Finally, Equation (47) follows from $\boldsymbol{X}_A \subseteq \boldsymbol{X}$, which implies that $I(\boldsymbol{X}; \boldsymbol{X}_A) = H(\boldsymbol{X}_A)$ and $I(\boldsymbol{X}; \boldsymbol{X}_A \mid Z_A) = H(\boldsymbol{X}_A \mid Z_A)$.

$\square$

## B.5 PROOF OF THEOREM 1

**Theorem 1.** *Each mixture-based multimodal VAE (Definition 3) approximates the expected log-evidence up to an irreducible discrepancy $\Delta(\boldsymbol{X}, \mathcal{S})$ that depends on the model-specific mixture distribution $\mathcal{S}$ as well as on the amount of modality-specific information in $\boldsymbol{X}$.*

*For the maximization of $\mathcal{L}_{\mathcal{S}}(\boldsymbol{x}; \theta, \phi)$ and every value of $\theta$ and $\phi$, the following inequality holds:*

$$\mathbb{E}_{p(\boldsymbol{x})}[\log p(\boldsymbol{x})] \geq \mathcal{L}_{\mathcal{S}}(\boldsymbol{x}; \theta, \phi) + \Delta(\boldsymbol{X}, \mathcal{S}) \tag{4}$$

*where*

$$\Delta(\boldsymbol{X}, \mathcal{S}) = \sum_{A \in \mathcal{S}} \omega_A \, H(\boldsymbol{X}_{\{1,\ldots,M\}\backslash A} \mid \boldsymbol{X}_A) \, . \tag{5}$$

*In particular, the generative discrepancy is always greater than or equal to zero and it is independent of $\theta$ and $\phi$ and thus remains constant during the optimization.*

*Proof.* Lemma 1 shows that all mixture-based multimodal VAEs approximate the expected log-evidence via the more general VIB objective

$$\min_{\psi} \sum_{A \in \mathcal{S}} \omega_A \left\{ H_{\psi}(\boldsymbol{X} \mid Z_A) + I_{\psi}(\boldsymbol{X}_A; Z_A) \right\} \tag{48}$$

where the encoding $Z_A = f_{\psi}(\boldsymbol{X}_A)$ is a function of a subset $\boldsymbol{X}_A \subseteq \boldsymbol{X}$.

The fact that $Z_A$ is a function of a *subset*, permits the following decomposition of the conditional entropy (see Lemma 2):

$$H(\boldsymbol{X} \mid Z_A) = H(\boldsymbol{X}_{\{1,\ldots,M\}\backslash A} \mid \boldsymbol{X}_A) + H(\boldsymbol{X}_A \mid Z_A) \, . \tag{49}$$

In particular, Equation (49) holds for every $Z_A = f_{\psi}(\boldsymbol{X}_A)$ and thus for every value $\psi$. Further, notice that $H(\boldsymbol{X}_{\{1,\ldots,M\}\backslash A} \mid \boldsymbol{X}_A)$ is independent of the learned encoding $Z_A$ and thus remains constant during the optimization with respect to $\psi$.

Hence, for every value $\psi$, the following inequality holds:

$$H_{\psi}(\boldsymbol{X} \mid Z_A) \geq H(\boldsymbol{X} \mid Z_A) \tag{50}$$

$$\geq H(\boldsymbol{X}_{\{1,\ldots,M\}\backslash A} \mid \boldsymbol{X}_A) \tag{51}$$

which means that the minimization of $H_{\psi}(\boldsymbol{X} \mid Z_A)$ is lower-bound by $H(\boldsymbol{X}_{\{1,\ldots,M\}\backslash A} \mid \boldsymbol{X}_A)$, even if $H_{\psi}(\boldsymbol{X} \mid Z_A)$ is a tight estimator of $H(\boldsymbol{X} \mid Z_A)$.

Analogously, for the optimization of the VIB objective (Equation (48)), for every value $\psi$, the following inequality holds:

$$\sum_{A \in \mathcal{S}} \omega_A \left\{ H_{\psi}(\boldsymbol{X} \mid Z_A) + I_{\psi}(\boldsymbol{X}_A; Z_A) \right\} \tag{52}$$

$$\geq \sum_{A \in \mathcal{S}} \omega_A \left\{ H(\boldsymbol{X} \mid Z_A) + I_{\psi}(\boldsymbol{X}_A; Z_A) \right\} \tag{53}$$

$$= \sum_{A \in \mathcal{S}} \omega_A \left\{ H(\boldsymbol{X}_A \mid Z_A) + I_{\psi}(\boldsymbol{X}_A; Z_A) \right\} + \underbrace{\sum_{A \in \mathcal{S}} \omega_A \, H(\boldsymbol{X}_{\{1,\ldots,M\}\backslash A} \mid \boldsymbol{X}_A)}_{\Delta(\boldsymbol{X}, \mathcal{S})} \tag{54}$$

where $\Delta(\boldsymbol{X}, \mathcal{S})$ is independent of $\psi$ and thus remains constant during the optimization. Consequently, $\Delta(\boldsymbol{X}, \mathcal{S})$ represents an irreducible error for the optimization of the VIB objective.

For mixture-based multimodal VAEs, Lemma 1 shows that $\mathcal{L}_{\mathcal{S}}(\boldsymbol{x}; \theta, \phi)$ is a special case of the VIB objective with $\psi = (\theta, \phi)$. Hence, for every value of $\theta$ and $\phi$, the following inequality holds:

$$\mathbb{E}_{p(\boldsymbol{x})}[\log p(\boldsymbol{x})] \geq \mathcal{L}_{\mathcal{S}}(\boldsymbol{x}; \theta, \phi) + \Delta(\boldsymbol{X}, \mathcal{S}) . \tag{55}$$

The exact value of $\Delta(\boldsymbol{X}, \mathcal{S})$ depends on the definition of the mixture distribution $\mathcal{S}$, as well as on the amount of modality-specific variation in the data. In particular, $\Delta(\boldsymbol{X}, \mathcal{S}) > 0$, if there is any subset $A \in \mathcal{S}$ with $\omega_A > 0$ for which $H(\boldsymbol{X}_{\{1,\ldots,M\}\setminus A} \mid \boldsymbol{X}_A) > 0$.

$\square$

### B.6 PROOF OF COROLLARY 1

**Corollary 1.** *Without modality sub-sampling, $\Delta(\boldsymbol{X}, \mathcal{S}) = 0$.*

*Proof.* Without modality sub-sampling, $\mathcal{S}$ is comprised of only one subset, the complete set of modalities $\{1, \ldots, M\}$, and therefore $\boldsymbol{X}_A = \boldsymbol{X}$ and $\boldsymbol{X}_{\{1,\ldots,M\}\setminus A} = \emptyset$. It follows that $\Delta(\boldsymbol{X}, \mathcal{S}) = H(\boldsymbol{X}_{\{1,\ldots,M\}\setminus A} \mid \boldsymbol{X}_A) = H(\emptyset \mid \boldsymbol{X}) = 0$, since the conditional entropy of the empty set is zero.

$\square$

### B.7 PROOF OF COROLLARY 2

**Corollary 2.** *For the MMVAE and MoPoE-VAE, the generative discrepancy increases given an additional modality $X_{M+1}$, if the new modality is sufficiently diverse in the following sense:*

$$\left(\frac{1}{|\mathcal{S}^+|} - \frac{1}{|\mathcal{S}|}\right) \sum_{A \in \mathcal{S}} I(\boldsymbol{X}_{\{1,\ldots,M\}\setminus A}; X_{M+1} \mid \boldsymbol{X}_A) < \frac{1}{|\mathcal{S}^+||\mathcal{S}|} \sum_{A \in \mathcal{S}} H(\boldsymbol{X}_A \mid X_{M+1}) + \tag{56}$$

$$\frac{1}{|\mathcal{S}^+|} \sum_{A \in \mathcal{S}} H(X_{M+1} \mid \boldsymbol{X}) \tag{57}$$

*where $\mathcal{S}$ denotes the model-specific mixture distribution over the set of subsets of modalities given modalities $X_1, \ldots, X_M$ and $\mathcal{S}^+$ is the respective mixture distribution over the extended set of subsets of modalities given $X_1, \ldots, X_{M+1}$.*

*Proof.* Let $X_{M+1}$ be the new modality, let $\boldsymbol{X}^+ := \{X_1, \ldots, X_{M+1}\}$ denote the extended set of modalities, and let $\mathcal{S}^+$ denote the new mixture distribution over subsets given $\boldsymbol{X}^+$. Note that all subsets from $\mathcal{S}$ are still contained in $\mathcal{S}^+$, but that $\mathcal{S}^+$ contains new subsets in addition to those in $\mathcal{S}$. Further, due to the re-weighting of mixture coefficients, $\mathcal{S}^+$ can have different mixture coefficients for the subsets it shares with $\mathcal{S}$. We denote by $S^- := \{(A, \omega_A^+) \in \mathcal{S}^+ : A \notin \mathcal{S}\}$ the set of new subsets and let $\omega_A^+$ denote the new mixture coefficients, where typically $\omega_A \neq \omega_A^+$ due to the re-weighting.

We are interested in the change of the generative discrepancy, when we add modality $X_{M+1}$:

$$\Delta(\boldsymbol{X}^+, \mathcal{S}^+) - \Delta(\boldsymbol{X}, \mathcal{S}) \tag{58}$$

$$= \sum_{B \in \mathcal{S}^+} \omega_B^+ H(\boldsymbol{X}_{\{1,\ldots,M+1\}\setminus B} \mid \boldsymbol{X}_B) - \sum_{A \in \mathcal{S}} \omega_A H(\boldsymbol{X}_{\{1,\ldots,M\}\setminus A} \mid \boldsymbol{X}_A) . \tag{59}$$

Re-write the right hand side in terms of subsets that are contained in both $\mathcal{S}$ and $\mathcal{S}^+$ and subsets that are only contained in $\mathcal{S}^+$. For this, we decompose the first term as follows

$$\sum_{B \in \mathcal{S}^+} \omega_B^+ H(\boldsymbol{X}_{\{1,\ldots,M+1\} \backslash B} \mid \boldsymbol{X}_B) \tag{60}$$

$$= \sum_{A \in \mathcal{S}} \omega_A^+ H(\boldsymbol{X}_{\{1,\ldots,M+1\} \backslash A} \mid \boldsymbol{X}_A) + \sum_{B \in \mathcal{S}^-} \omega_B^+ H(\boldsymbol{X}_{\{1,\ldots,M+1\} \backslash B} \mid \boldsymbol{X}_B) \tag{61}$$

$$= \sum_{A \in \mathcal{S}} \omega_A^+ H(\boldsymbol{X}_{\{1,\ldots,M\} \backslash A} \mid \boldsymbol{X}_A) + \sum_{A \in \mathcal{S}} \omega_A^+ H(X_{M+1} \mid \boldsymbol{X}) + \tag{62}$$

$$\sum_{B \in \mathcal{S}^-} \omega_B^+ H(\boldsymbol{X}_{\{1,\ldots,M+1\} \backslash B} \mid \boldsymbol{X}_B) \tag{63}$$

where the last equation follows from

$$H(\boldsymbol{X}_{\{1,\ldots,M+1\} \backslash A} \mid \boldsymbol{X}_A) = H(\boldsymbol{X}_{\{1,\ldots,M\} \backslash A} \mid \boldsymbol{X}_A) + H(X_{M+1} \mid \boldsymbol{X}_A, \boldsymbol{X}_{\{1,\ldots,M\} \backslash A}) \tag{64}$$

$$= H(\boldsymbol{X}_{\{1,\ldots,M\} \backslash A} \mid \boldsymbol{X}_A) + H(X_{M+1} \mid \boldsymbol{X}) . \tag{65}$$

We can use the decomposition from Equation (63) to re-write the right hand side of Equation (59) by collecting the corresponding terms for $H(\boldsymbol{X}_{\{1,\ldots,M\} \backslash A} \mid \boldsymbol{X}_A)$:

$$\sum_{A \in \mathcal{S}} (\omega_A^+ - \omega_A) H(\boldsymbol{X}_{\{1,\ldots,M\} \backslash A} \mid \boldsymbol{X}_A) + \sum_{A \in \mathcal{S}} \omega_A^+ H(X_{M+1} \mid \boldsymbol{X}) + $$
$$\sum_{B \in \mathcal{S}^-} \omega_B^+ H(\boldsymbol{X}_{\{1,\ldots,M+1\} \backslash B} \mid \boldsymbol{X}_B) . \tag{66}$$

Notice that in Equation (66) only the first term can be negative, due to the re-weighting of mixture coefficients for terms that do not contain $X_{M+1}$. Hence, in the general case, the generative discrepancy can only decrease, if the mixture coefficients change in such a way that the first term in Equation (66) dominates the other two terms.

For the relevant special case of uniform mixture weights, which applies to both the MMVAE and MoPoE-VAE, we can further decompose Equation (66) into $(i)$ information shared between $\boldsymbol{X}$ and $X_{M+1}$, and $(ii)$ information that is specific to $\boldsymbol{X}$ or $X_{M+1}$.

Using uniform mixture coefficients $\omega_A = \frac{1}{|\mathcal{S}|}$ and $\omega_A^+ = \frac{1}{|\mathcal{S}^+|}$ for all subsets, we can factor out the coefficients and re-write Equation (66) as follows:

$$\left( \frac{1}{|\mathcal{S}^+|} - \frac{1}{|\mathcal{S}|} \right) \sum_{A \in \mathcal{S}} H(\boldsymbol{X}_{\{1,\ldots,M\} \backslash A} \mid \boldsymbol{X}_A) + \frac{1}{|\mathcal{S}^+|} \sum_{A \in \mathcal{S}} H(X_{M+1} \mid \boldsymbol{X}) + $$
$$\frac{1}{|\mathcal{S}^+|} \sum_{B \in \mathcal{S}^-} H(\boldsymbol{X}_{\{1,\ldots,M+1\} \backslash B} \mid \boldsymbol{X}_B) \tag{67}$$

where the second term already denotes information that is specific to $X_{M+1}$. Hence, we decompose the first and last terms corresponding to $(i)$ and $(ii)$.

For the first term from Equation (67), we have

$$\left( \frac{1}{|\mathcal{S}^+|} - \frac{1}{|\mathcal{S}|} \right) \sum_{A \in \mathcal{S}} H(\boldsymbol{X}_{\{1,\ldots,M\} \backslash A} \mid \boldsymbol{X}_A) \tag{68}$$

$$= \left( \frac{1}{|\mathcal{S}^+|} - \frac{1}{|\mathcal{S}|} \right) \sum_{A \in \mathcal{S}} \left\{ H(\boldsymbol{X}_{\{1,\ldots,M\} \backslash A} \mid \boldsymbol{X}_A, X_{M+1}) + I(\boldsymbol{X}_{\{1,\ldots,M\} \backslash A}; X_{M+1} \mid \boldsymbol{X}_A) \right\}. \tag{69}$$

For the last term from Equation (67), we have

$$\frac{1}{|\mathcal{S}^+|} \sum_{B \in \mathcal{S}^-} H(\boldsymbol{X}_{\{1,\ldots,M+1\} \backslash B} \mid \boldsymbol{X}_B) \tag{70}$$

$$= \frac{1}{|\mathcal{S}^+|} \left\{ H(\boldsymbol{X} \mid X_{M+1}) + \sum_{A \in \mathcal{S}} \mathbf{1}_{\{(A \cup \{M+1\}) \in \mathcal{S}^-\}} H(\boldsymbol{X}_{\{1,\ldots,M\} \backslash A} \mid \boldsymbol{X}_A, X_{M+1}) \right\} \tag{71}$$

where we can further decompose

$$\frac{1}{|\mathcal{S}^+|} H(\boldsymbol{X} \mid X_{M+1}) = \frac{1}{|\mathcal{S}^+|} \Big\{ H(\boldsymbol{X} \mid \boldsymbol{X}_A, X_{M+1}) + I(\boldsymbol{X}; \boldsymbol{X}_A \mid X_{M+1}) \Big\} \tag{72}$$

$$= \frac{1}{|\mathcal{S}^+|} \Big\{ H(\boldsymbol{X} \mid \boldsymbol{X}_A, X_{M+1}) + H(\boldsymbol{X}_A \mid X_{M+1}) \Big\} \tag{73}$$

$$= \frac{1}{|\mathcal{S}^+||\mathcal{S}|} \sum_{A \in \mathcal{S}} \Big\{ H(\boldsymbol{X}_{\{1,\dots,M\}\backslash A} \mid \boldsymbol{X}_A, X_{M+1}) + H(\boldsymbol{X}_A \mid X_{M+1}) \Big\}. \tag{74}$$

Collecting all corresponding terms from Equations (69), (71) and (74), we can re-write Equation (67) as follows:

$$\left( \frac{1}{|\mathcal{S}^+|} - \frac{1}{|\mathcal{S}|} + \frac{1}{|\mathcal{S}^+||\mathcal{S}|} \right) \sum_{A \in \mathcal{S}} H(\boldsymbol{X}_{\{1,\dots,M\}\backslash A} \mid \boldsymbol{X}_A, X_{M+1}) + \tag{75}$$

$$\left( \frac{1}{|\mathcal{S}^+|} - \frac{1}{|\mathcal{S}|} \right) \sum_{A \in \mathcal{S}} I(\boldsymbol{X}_{\{1,\dots,M\}\backslash A}; X_{M+1} \mid \boldsymbol{X}_A) + \tag{76}$$

$$\frac{1}{|\mathcal{S}^+|} \sum_{A \in \mathcal{S}} \mathbf{1}_{\{(A \cup \{M+1\}) \in \mathcal{S}^-\}} H(\boldsymbol{X}_{\{1,\dots,M\}\backslash A} \mid \boldsymbol{X}_A, X_{M+1}) + \tag{77}$$

$$\frac{1}{|\mathcal{S}^+||\mathcal{S}|} \sum_{A \in \mathcal{S}} H(\boldsymbol{X}_A \mid X_{M+1}) + \tag{78}$$

$$\frac{1}{|\mathcal{S}^+|} \sum_{A \in \mathcal{S}} H(X_{M+1} \mid \boldsymbol{X}). \tag{79}$$

For both the MMVAE and MoPoE, the first and last terms cancel out, which can see by plugging in the respective definitions of $\mathcal{S}$ into the above equation. Recall that for the MMVAE, $\mathcal{S}$ is comprised of the set of unimodal subsets $\{\{\boldsymbol{x}_1\}, \dots, \{\boldsymbol{x}_M\}\}$ and thus $\mathcal{S}^+$ is comprised of $\{\{\boldsymbol{x}_1\}, \dots, \{\boldsymbol{x}_{M+1}\}\}$. For the MoPoE-VAE, $\mathcal{S}$ is comprised of the powerset $\mathcal{P}(M) \setminus \{\emptyset\}$ and thus $\mathcal{S}^+$ is comprised of the powerset $\mathcal{P}(M+1) \setminus \{\emptyset\}$. Hence, for the MMVAE and MoPoE-VAE, we have shown that $\Delta(\boldsymbol{X}^+, \mathcal{S}^+) - \Delta(\boldsymbol{X}, \mathcal{S})$ is equal to the following expression:

$$\left( \frac{1}{|\mathcal{S}^+|} - \frac{1}{|\mathcal{S}|} \right) \sum_{A \in \mathcal{S}} I(\boldsymbol{X}_{\{1,\dots,M\}\backslash A}; X_{M+1} \mid \boldsymbol{X}_A) + \tag{80}$$

$$\frac{1}{|\mathcal{S}^+||\mathcal{S}|} \sum_{A \in \mathcal{S}} H(\boldsymbol{X}_A \mid X_{M+1}) + \frac{1}{|\mathcal{S}^+|} \sum_{A \in \mathcal{S}} H(X_{M+1} \mid \boldsymbol{X}) \tag{81}$$

where the information is decomposed into:

$(i)$ information shared between $\boldsymbol{X}$ and $X_{M+1}$ (term (80)), and

$(ii)$ information that is specific to $\boldsymbol{X}$ or $X_{M+1}$ (the first and second terms in (81) respectively),

and where only $(i)$ can be negative since $|\mathcal{S}^+| > |\mathcal{S}|$. This concludes the proof of Corollary 2, showing that $\Delta(\boldsymbol{X}^+, \mathcal{S}^+) - \Delta(\boldsymbol{X}, \mathcal{S}) > 0$, if $X_{M+1}$ is sufficiently diverse in the sense that $(ii) > (i)$.

$\square$

# C  EXPERIMENTS

## C.1  DESCRIPTION OF THE DATASETS

**PolyMNIST**   The PolyMNIST dataset, introduced in Sutter et al. (2021), combines the MNIST dataset (LeCun et al., 1998) with crops from five different background images to create five synthetic image modalities. Each sample from the data is a set of five MNIST images (with digits of the same class) overlayed on $28 \times 28$ crops from five different background images. Figure 1a shows 10 samples from the PolyMNIST dataset; each column represents one sample and each row represents one modality. The dataset provides a convenient testbed for the evaluation of generative coherence, because by design only the digit information is shared between modalities.

**Translated-PolyMNIST**   This new dataset is conceptually similar to PolyMNIST in that a digit label is shared between five synthetic image modalities. The difference is that in the creation of the dataset, we change the size and position of the digit, as shown in Figure 1b. Technically, instead of overlaying a full-sized $28 \times 28$ MNIST digit on a patch from the respective background image, we downsample the MNIST digit by a factor of two and place it at a random $(x, y)$-coordinate within the $28 \times 28$ background patch. Conceptually, these transformations leave the shared information between modalities (i.e., the digit label) unaffected and only serve to make it more difficult to predict the shared information across modalities on expectation.

**Caltech Birds (CUB)**   The extended CUB dataset from Shi et al. (2019) is comprised of two modalities, images and captions. Each image from Caltech-Birds (CUB-200-2011 Wah et al., 2011) is coupled with 10 crowdsourced descriptions of the respective bird. Figure 1c shows five samples from the dataset. It is important to note that we use the CUB dataset with *real images*, instead of the simplified version based on precomputed ResNet-features that was used in Shi et al. (2019; 2021).

## C.2  IMPLEMENTATION DETAILS

Our experiments are based on the publicly available code from Sutter et al. (2021), which already provides an implementation of PolyMNIST. A notable difference in our implementation is that we employ ResNet architectures, because we found that the previously used convolutional neural networks did not have sufficient capacity for the more complex datasets we use. For internal consistency, we use ResNets for PolyMNIST as well. We have verified that there is no significant difference compared to the results from Sutter et al. (2021) when we change to ResNets.

**Hyperparameters**   All models were trained using the Adam optimizer (Kingma and Ba, 2015) with learning rate 5e-4 and a batch size of 256. For image modalities we estimate likelihoods using Laplace distributions and for captions we employ one-hot categorical distributions. Models were trained for 500, 1000, and 150 epochs on PolyMNIST, Translated-PolyMNIST, and CUB respectively. Similar to previous work, we use Gaussian priors and a latent space with 512 dimensions for PolyMNIST and 64 dimensions for CUB. For a fair comparison, we reduce the latent dimensionality of unimodal VAEs proportionally (wrt. the number of modalities) to control for capacity. For the $\beta$-ablations, we use $\beta \in \{$3e-4, 3e-3, 3e-1, 1, 3, 9$\}$ and, in addition, 32 for CUB.

**Evaluation metrics**   For the evaluation of *generative quality*, we use the Fréchet inception distance (FID; Heusel et al., 2017), a standard metric for evaluating the quality of generated images. In Appendix C.3, we also provide log-likelihoods and qualitative results for both images and captions. To compute *generative coherence*, we adopt the definitions from previous works (Shi et al., 2019; Sutter et al., 2021). Generative coherence requires annotation on what is shared between modalities; for example, in both PolyMNIST and Translated-PolyMNIST the digit label is shared by design. For a single generated example $\hat{\boldsymbol{x}}_m \sim q_\phi(\boldsymbol{x}_m \mid \boldsymbol{z})$ from modality $m$, the generative coherence is computed as the following indicator:

$$\text{Coherence}(\hat{\boldsymbol{x}}_m, y, g_m) = \mathbf{1}_{\{g_m(\hat{\boldsymbol{x}}_m) = y\}} \tag{82}$$

where $y$ is a ground-truth class label and $g_m$ is a pretrained classifier (learned on the training data from modality $m$) that outputs a predicted class label. To compute the *conditional coherence accuracy*, we average the coherence values over a set of $N$ conditionally generated examples, where $N$ is

typically the size of the test set. In particular, when $\hat{\boldsymbol{x}}_m \sim q_\phi(\boldsymbol{x}_m \mid \boldsymbol{z})$ is conditionally generated from $\boldsymbol{z} \sim p_\theta(\boldsymbol{z} \mid \boldsymbol{x}_A)$ such that $A = \{1, \ldots, M\} \setminus m$, the metric is specified as the *leave-one-out conditional coherence accuracy*, because the input consists of all modalities except the one that is being generated. When it is clear from context which metric is used, we refer to the (leave-one-out) conditional coherence accuracy simply as generative coherence. For PolyMNIST, we use the pretrained digit classifiers that are provided in the publicly available code from Sutter et al. (2021) and for Translated-PolyMNIST we train the classifiers from scratch with the same architectures that are used for the VAE encoders. Notably, the new pretrained digit classifiers have a classification accuracy between 93.5–96.9% on the test set of the respective modality, which means that it is possible to predict the digits fairly well with the given architectures.

### C.3 Additional experimental results

**Linear classification**    Shi et al. (2019) propose linear classification as a measure of latent factorization, to judge the quality of learned representations and to assess how well the information decomposes into shared and modality-specific features. Figure 6 shows the linear classification accuracy on the learned representations. The results suggest that not only does the generative coherence decline when we switch from PolyMNIST to Translated-PolyMNIST, but also the quality of the learned representations. While a low classification accuracy does not imply that there is no digit information encoded in the latent representation (after all, digits show up in most self-reconstructions), the result demonstrates that a *linear* classifier cannot extract the digit information.

**Log-likelihoods and qualitative results**    Figure 7 shows the generative quality in terms of joint log-likelihoods. We observe a similar ranking of models as with FID, but we notice that the gap between MVAE and MoPoE-VAE appears less pronounced. The reason for this discrepancy is that, to be consistent with Sutter et al. (2021), we estimate joint log-likelihoods given *all* modalities—a procedure that resembles reconstruction more than it does unconditional generation. It can be of independent interest that log-likelihoods might overestimate the generative quality for unconditional generation for certain types of models. Qualitative results for unconditional generation (Figure 9) support the hypothesis that the presented log-likelihoods do not reflect the visible lack of generative quality for the MoPoE-VAE. Further, qualitative results for conditional generation (Figure 10) indicate a lack of diversity for both the MMVAE and MoPoE-VAE: even though we draw different samples from the posterior, the respective conditionally generated samples (i.e., the ten samples along each column) show little diversity in terms of backgrounds or writing styles.

**Repeated modalities**    To check if the generative quality gap is also present when modalities have *similar* modality-specific variation, we use PolyMNIST with "repeated" modalities generated from the same background image (illustrated in Figure 5). We vary the number of modalities from 2 to 5, but in contrast to the results from Figure 3, we now use repeated modalities. Figure 11 confirms that the generative quality of both the MVAE and MoPoE-VAE deteriorates with each additional modality, even in this simplified setting with repeated modalities. In comparison, the generative quality of the MVAE is much closer to the unimodal VAE for any number of modalities. These results lend further support to the theoretical statements from Corollaries 1 and 2.

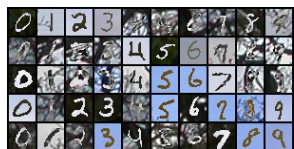

Figure 5: PolyMNIST with five "repeated" modalities.

**MMVAE with the official implementation**    The empirical results of the MMVAE in Section 5 are based on a simplified version of the model that was proposed by Shi et al. (2019). In particular, we use the re-implementation from Sutter et al. (2021), which optimizes the standard ELBO and not the doubly reparameterized ELBO gradient estimator (DReG, Tucker et al., 2019) with importance sampling that is used in the official implementation from Shi et al. (2019). Further, the re-implementation does not parameterize the prior but uses a fixed, standard normal prior instead.

To verify that these implementation differences do not affect the core results—the generative quality gap and the lack of coherence—we conducted experiments using the MMVAE with the official implementation from Shi et al. (2019). Figure 12 shows the $\beta$-ablation for PolyMNIST and it confirms that there is still a clear gap in generative quality between the unimodal VAE and the MMVAE when we use the official implementation. For Translated-PolyMNIST (not shown) the

results are similar; in particular, we have verified that generative coherence for cross generation is random, even if we limit the dataset to two modalities.

**MVAE with ELBO sub-sampling**   For the MVAE, Wu and Goodman (2018) introduce ELBO sub-sampling as an additional training strategy to learn the inference networks for different subsets of modalities. In our notation, ELBO sub-sampling can be described by the following objective:

$$\mathcal{L}(\boldsymbol{x}; \theta, \phi) + \sum_{A \in \mathcal{S}} \mathcal{L}(\boldsymbol{x}_A; \theta, \phi) \tag{83}$$

where $\mathcal{S}$ denotes some set of subsets of modalities. Wu and Goodman (2018) experiment with different choices for $\mathcal{S}$, but throughout all of their experiments they use at least the set of unimodal subsets $\{\{\boldsymbol{x}_1\}, \ldots, \{\boldsymbol{x}_M\}\}$, which yields the following objective:

$$\mathcal{L}(\boldsymbol{x}; \theta, \phi) + \sum_{i=1}^{M} \mathcal{L}(\boldsymbol{x}_i; \theta, \phi) . \tag{84}$$

It is important to note that the above objective differs from the objective optimized by all mixture-based multimodal VAEs (Definition 3) in that there are no cross-modal reconstructions in Equation (84). As a consequence, ELBO sub-sampling puts more weight on the approximation of the marginal distributions compared to the conditionals and therefore does not optimize a proper bound on the joint distribution (Wu and Goodman, 2019).

Figure 13 shows the PolyMNIST $\beta$-ablation comparing MVAE with and without ELBO sub-sampling. MVAE$^+$ denotes the model with ELBO sub-sampling using objective (84). Notably, MVAE$^+$ achieves significantly better generative coherence, while both models perform similarly in terms of generative quality (both in terms of FID and joint log-likelihood). Hence, even though the MVAE$^+$ optimizes an incorrect bound on the joint distribution (Wu and Goodman, 2019), our results suggest that the learned models behave quite similar in practice, which can be of independent interest for future work.

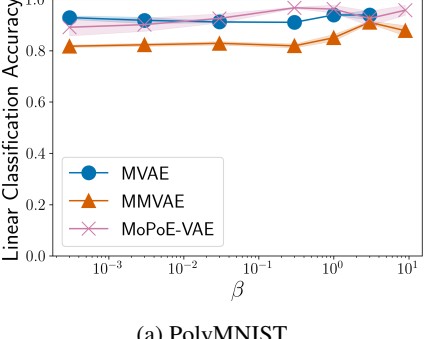

(a) PolyMNIST

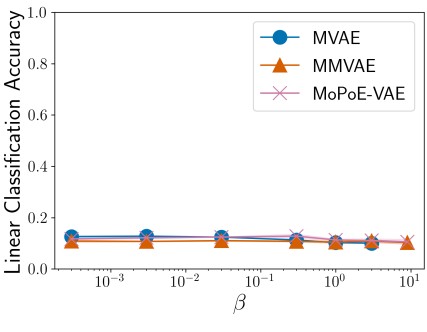

(b) Translated-PolyMNIST

Figure 6: Linear classification of latent representations. For each model, linear classifiers were trained on the joint embeddings from 500 randomly sampled training examples. Points denote the average digit classification accuracy of the respective classifiers. The results are averaged over three seeds and the bands show one standard deviation respectively. Due to numerical instabilities, the MVAE could not be trained with larger $\beta$ values. For CUB, classification performance cannot be computed, because shared factors are not annotated.

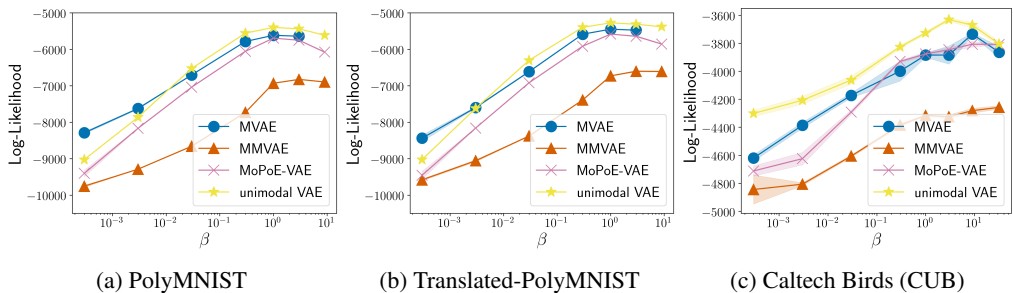

Figure 7: Joint log-likelihoods over a range of $\beta$ values. Each point denotes the estimated joint log-likelihood averaged over three different seeds and the bands show one standard deviation respectively. Due to numerical instabilities, the MVAE could not be trained with larger $\beta$ values.

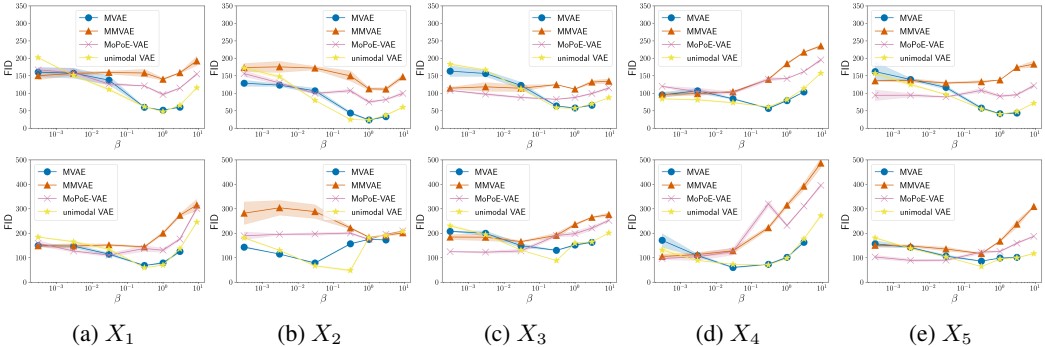

Figure 8: FID for modalities $X_1, \ldots, X_5$. The top row shows all FIDs for PolyMNIST and the bottom row for Translated-PolyMNIST respectively. Points denote the FID averaged over three seeds and bands show one standard deviation respectively. Due to numerical instabilities, the MVAE could not be trained with larger $\beta$ values.

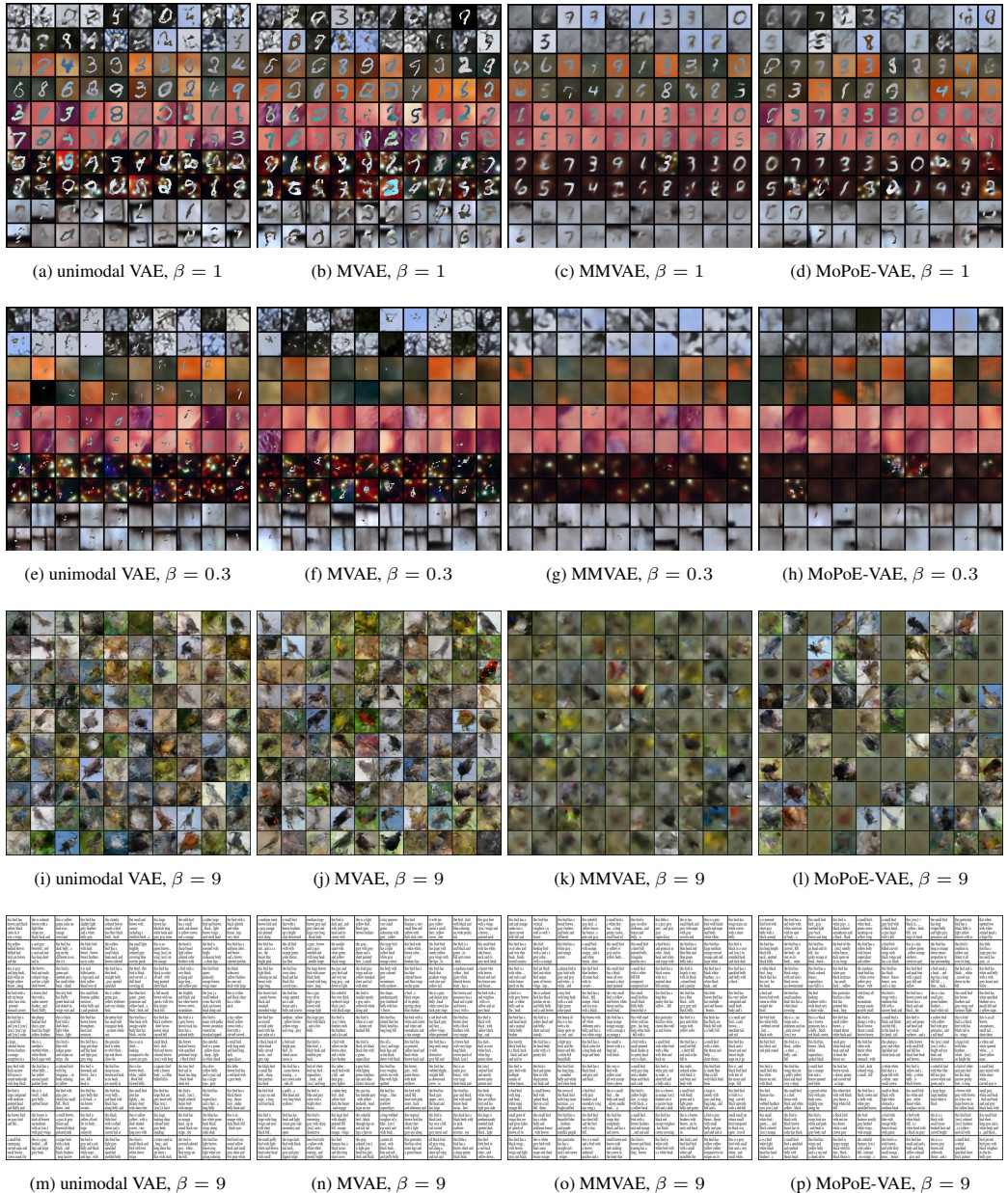

Figure 9: Qualitative results for the unconditional generation using prior samples. For PolyMNIST (Subfigures (a) to (d)) and Translated-PolyMNIST (Subfigures (e) to (h)), we show 20 samples for each modality. For CUB, we show 100 generated images (Subfigures (i) to (l)) and 100 generated captions (Subfigures (m) to (p)) respectively. Best viewed zoomed and in color.

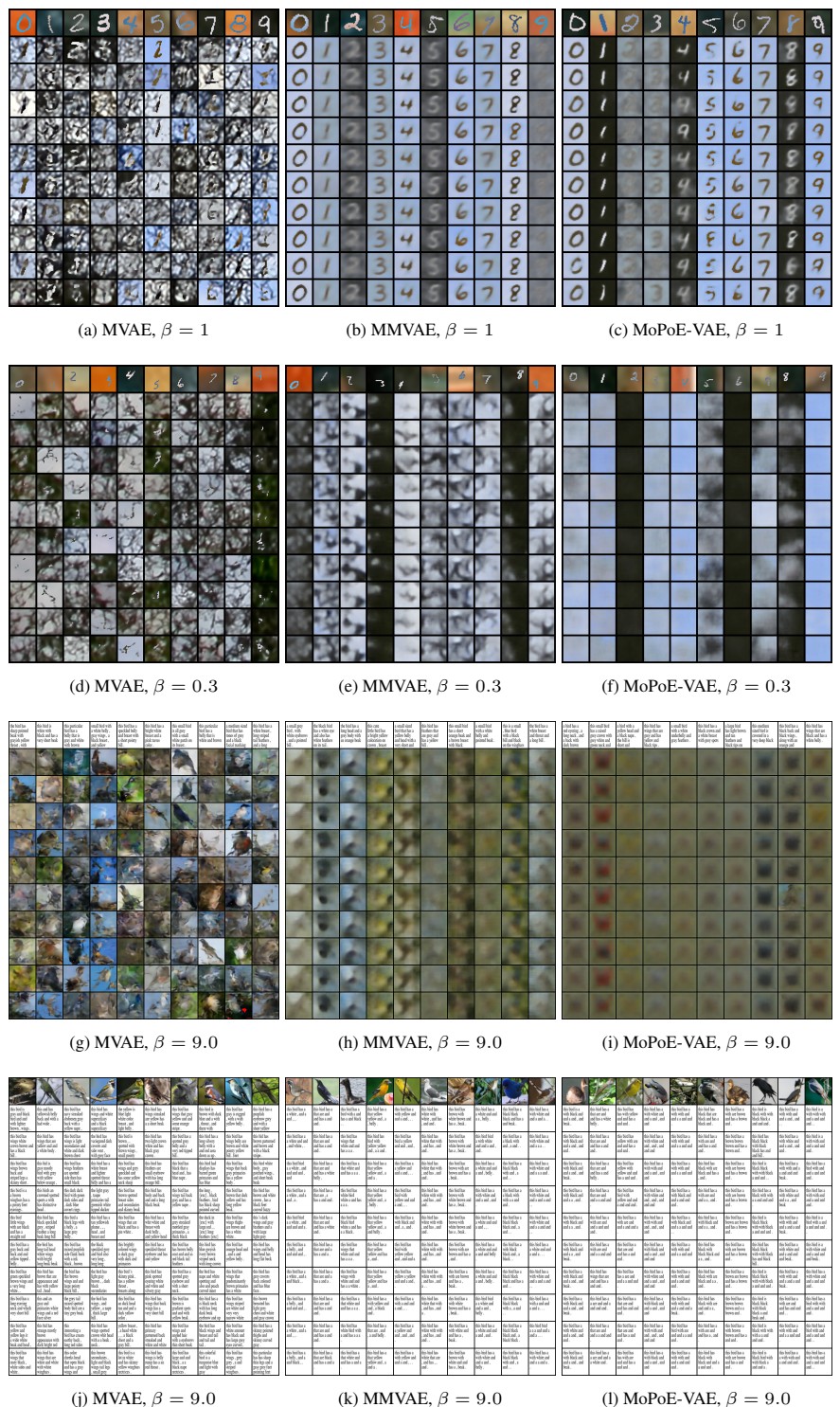

(a) MVAE, $\beta = 1$     (b) MMVAE, $\beta = 1$     (c) MoPoE-VAE, $\beta = 1$

(d) MVAE, $\beta = 0.3$     (e) MMVAE, $\beta = 0.3$     (f) MoPoE-VAE, $\beta = 0.3$

(g) MVAE, $\beta = 9.0$     (h) MMVAE, $\beta = 9.0$     (i) MoPoE-VAE, $\beta = 9.0$

(j) MVAE, $\beta = 9.0$     (k) MMVAE, $\beta = 9.0$     (l) MoPoE-VAE, $\beta = 9.0$

Figure 10: Qualitative results for the conditional generation across modalities. For PolyMNIST (Subfigures (a) to (c)) and Translated-PolyMNIST (Subfigures (d) to (f)), we show 10 conditionally generated samples of modality $X_1$ given the sample from modality $X_2$ that is shown in the first row of the respective subfigure. For CUB, we show the generation of images given captions (Subfigures (g) to (i)), as well as the generation of captions given images (Subfigures (j) to (l)). Best viewed zoomed and in color.

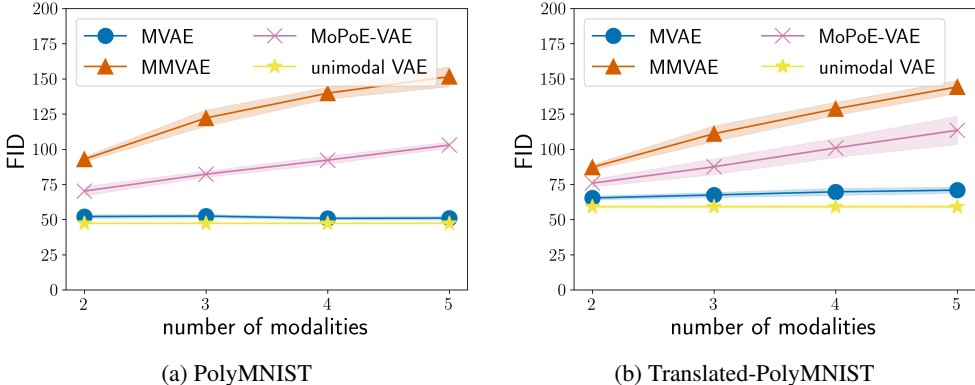

(a) PolyMNIST

(b) Translated-PolyMNIST

Figure 11: Generative quality as a function of the number of modalities. In contrast to Figure 3, here we "repeat" the same modality, to verify that the generative quality also declines when the modality-specific variation of all modalities is similar. All models are trained with $\beta = 1$ on PolyMNIST and $\beta = 0.3$ on Translated-PolyMNIST. The results are averaged over three seeds and all modalities; the bands show one standard deviation respectively. For the unimodal VAE, which uses only a single modality, the average and standard deviation are plotted as a constant.

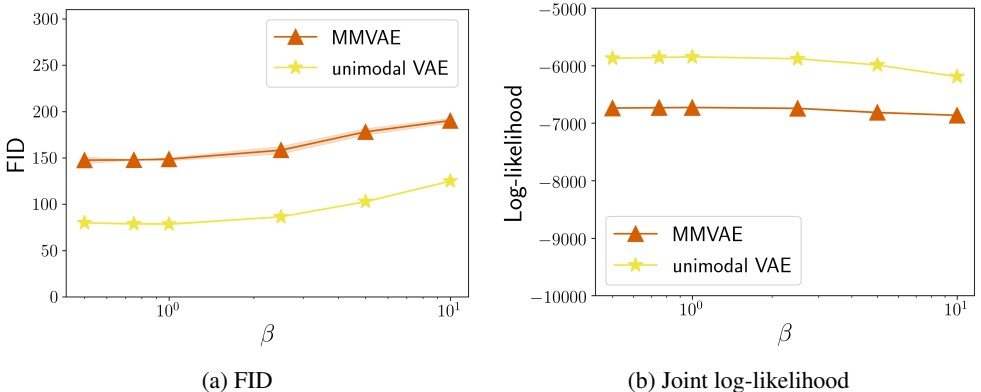

(a) FID

(b) Joint log-likelihood

Figure 12: PolyMNIST $\beta$-ablation using the official implementation of the MMVAE. In particular, for both the MMVAE and the unimodal VAE, we use the DReG objective, importance sampling, as well as a learned prior. Points denote the value of the respective metric averaged over three seeds and bands show one standard deviation respectively.

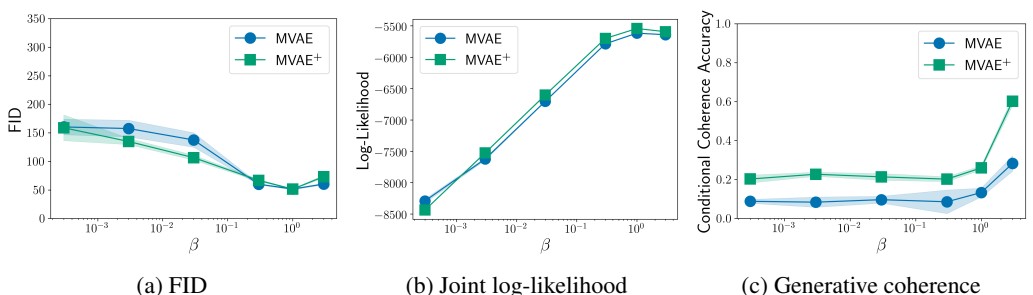

(a) FID

(b) Joint log-likelihood

(c) Generative coherence

Figure 13: PolyMNIST $\beta$-ablation, comparing MVAE with and without additional ELBO sub-sampling. MVAE$^+$ denotes the model with additional ELBO sub-sampling. Points denote the value of the respective metric averaged over three seeds and bands show one standard deviation respectively.

