# OpenReview forum: "On the Limitations of Multimodal VAEs"
_ICLR.cc/2022/Conference — ICLR 2022 Poster_

### Official Review · Reviewer_B728 · 2021-11-01

**Correctness:** 3
**Technical Novelty And Significance:** 4
**Empirical Novelty And Significance:** 3
**Recommendation:** 6
**Confidence:** 3

**Main Review:**

Overall, the paper is clearly written and easy to follow. The topic of this paper is very interesting and could have some implications in the area of VAE, or more broadly, generative models.

The strengths of this paper include the following.
- The writing in most parts of the paper is clear.
- The theory presented in section 4 is concise but very informative.
- The experiments are extensive.

The weakness of this paper includes the following.
- As an addition to Thm 1, it is useful to give an example that compares the ELBO and the gap $\Delta$. This should tell us which term dominates. If the ELBO dominates then your theory may not be able to explain the quality drop.
- As for the claim that the gap $\Delta$ degrades quality of multimodal VAEs compared to unimodal VAEs, I think the authors should have a paragraph discussing the following. Are you assuming that a larger gap $\Delta$ leads to worse quality? However, optimizing the ELBO can sometimes be better than optimizing the likelihood (e.g. VAE sometimes outperforms likelihood-based models such as normalizing flows), especially when the model has limited capacity - as in this case, likelihood-based models suffer from memorizing a huge amount of modes. From this view, the gap between likelihood and ELBO (for unimodal VAE) may have good inductive bias. So, could the $\Delta$ in Thm 1 have any good inductive bias?
- Following the above point, it is also valuable to conduct an experiment where you gradually increase $\Delta$ and see if the quality monotonically drops.
- There is no theory for coherence. In section 5.2, the description about this concept is not very clear. What does it mean by saying "a model's ability to generate semantically related samples across modalities"? Also, a formal, mathematical definition for the leave-one-out coherence also helps understanding of this part.

Some minors:
- Eq (1): $\sum_{A\in\mathcal{S}}$ is not precise because you are in fact summing over $(A,\omega_A)$.
- Figures should be made larger for better readability.

The evaluation of this paper is based on the above points. The score may be modified based on feedback and updates from authors.

**Summary Of The Paper:**

The paper looks at the ELBO loss function for multimodal VAE, and theoretically proves a nontrivial gap $\Delta(X,S)$ between the likelihood and ELBO, from an information theory point of view. The paper claims this gap leads to quality drop of multimodal VAEs. The paper then conducts experiments and empirically examines the tradeoff between unimodal and multimodal VAEs.

**Summary Of The Review:**

Theory is concise but informative, and experiments are extensive. However, there are a few key things missed from the paper, as discussed in the main review.

--------------------
Update after rebuttal

I have read the response and I will keep my score. It is a very interesting paper. I encourage the authors to add some of the discussions to either the main text or the appendix.

---

> ### Author Response · Authors · 2021-11-19
> **reply to reviewer B728 (part 1/2)**
>
> > As an addition to Thm 1, it is useful to give an example that compares the ELBO and the gap $\Delta$. This should tell us which term dominates. If the ELBO dominates then your theory may not be able to explain the quality drop.
>
> First, we would like to point out that $E_{p(\boldsymbol{x})}[\log p(\boldsymbol{x})]  \geq E_{p(\boldsymbol{x})}[\log p(\boldsymbol{x})] - \Delta(\boldsymbol{X}, \mathcal{S}) \geq ELBO_{\mathcal{S}}(\boldsymbol{X})$. Hence, the ELBO can, at best, approximate $E_{p(\boldsymbol{x})}[\log p(\boldsymbol{x})] - \Delta(\boldsymbol{X}, \mathcal{S})$, where $\Delta(\boldsymbol{X}, \mathcal{S}) \geq 0$. Given a particular model (e.g., the MMVAE or MoPoE-VAE), the exact size of $\Delta$ depends only on the data. Hence, only if there is very little modality-specific information in *all* modalities, we have $\Delta \rightarrow 0$ for models that sub-sample modalities. Note that this condition requires modalities to be extremely similar, which does not apply to most multimodal datasets, where $\Delta$ (i.e., the amount of modality-specific variation) typically represents a large part of the total variation in the data. We have extended the paragraph below Corollary 2 with this example.
>
> > As for the claim that the gap degrades quality of multimodal VAEs compared to unimodal VAEs, I think the authors should have a paragraph discussing the following. Are you assuming that a larger gap leads to worse quality?
>
>
> To be precise, we do not suggest a strict directionality of the form “larger $\Delta$ implies worse (generative) quality”. However, we do argue that there is a *negative association* between $\Delta$ and the generative quality, and our empirical results confirm that such an association can be observed across all of the considered datasets. For example, in Figure 2 we observe that across all of the considered datasets the generative quality of the MVAE (which has $\Delta = 0$) is significantly better than the generative quality of the MMVAE and MoPoE-VAE (both of which have $\Delta > 0$). To clarify the point that you have raised, in the beginning of Section 5.1 we made it more explicit that we expect to see a negative association between $\Delta$ and the generative quality.
>
> > Following the above point, it is also valuable to conduct an experiment where you gradually increase $\Delta$ and see if the quality monotonically drops.
>
> Thank you for this helpful suggestion. We already showcase the negative association between $\Delta$ and the generative quality in two experiments. In Figure 2, we vary $\Delta$ by comparing different models and show that models with a smaller $\Delta$ exhibit a better generative quality across all of the considered datasets. In Figures 3 and 11, we gradually increase $\Delta$ by increasing the number of modalities to show a monotonic decrease in the generative quality of the MMVAE and MoPoE-VAE (both of which have $\Delta > 0$). It would also be interesting to design an experiment where $\Delta$ can be measured exactly and where it is increased by an adaptation of the dataset in a way that increases only the modality-specific variation in the data. We think that this would be possible to test on simulated data and we have pointed this out as an opportunity for future work (Section 6).
>
>
> > So, could the $\Delta$ in Thm 1 have any good inductive bias?
>
> Thanks for the great question and yes, this is indeed possible. One can already observe an inductive bias of sub-sampling in our experiments. In particular, in Figures 2a and 4a we see the following tradeoff between two different metrics for generative performance (generative quality and generative coherence): while the MVAE shows significantly better generative quality, its generative coherence on PolyMNIST is significantly worse compared to the MMVAE and MoPoE-VAE. This tradeoff suggests that modality sub-sampling might have a good inductive bias for datasets where shared information can be predicted in expectation across modalities. We have clarified the possibility of an inductive bias in Section 4.3 and Section 5.1.
>
> > There is no theory for coherence. In section 5.2, the description about this concept is not very clear. What does it mean by saying "a model's ability to generate semantically related samples across modalities"? Also, a formal, mathematical definition for the leave-one-out coherence also helps understanding of this part.
>
>
> Thank you for pointing this out. We adopted the definition from previous work (Shi et al., 2019; Sutter et al. 2021), but to be more precise, we have now added a formal definition of the computed quantities to the Appendix (Section C.2; paragraph on evaluation metrics).
>
> $ $
>
> *[NOTE: this reply continues in a separate comment ]*

---

> ### Author Response · Authors · 2021-11-19
> **reply to reviewer B728 (part 2/2)**
>
> > Eq (1): $\sum_{A \in \mathcal{S}}$ is not precise because you are in fact summing over $(A, \omega_A)$.
>
> Thanks! We now introduce the use of our abbreviated notation below Definition 2.
>
> > Figures should be made larger for better readability.
>
> We have increased all figures as much as the remaining space permits us to. In particular, for better readability, we have increased the fontsize of all labels and the size of all markers.
>
> > Theory is concise but informative, and experiments are extensive. However, there are a few key things missed from the paper, as discussed in the main review.
>
> Thank you for the overall positive feedback! We hope that our response clarifies the remaining issues.

---

### Official Review · Reviewer_UGYa · 2021-11-02

**Correctness:** 3
**Technical Novelty And Significance:** 3
**Empirical Novelty And Significance:** 4
**Recommendation:** 8
**Confidence:** 5

**Main Review:**

Strengths:
- The paper is very well organized and easy to read.
- It theoretically shows a fundamental problem in mixture-based multimodal VAEs, which has not been pointed out before. This problem is intuitively convincing.
- The experiments using more complex datasets clearly support the theoretical results of this study. These results are clearly novel and important for future research in multimodal VAEs.

Weaknesses:
- The authors define models with Equation 2 as the objective as mixture-based multimodal VAEs, and they show the limitations of Equation 2 due to sub-sampling. However, this objective is a lower bound of the "true" objectives of MMVAE and MoPoE-VAE, as pointed out by the authors. Therefore, it is unclear whether the problems pointed out in this paper occur only in Equation 2 or also in these true objectives (I understand that the actual implementation in [Sutter+ 2021] uses Equation 2, but my point is that the authors should clarify the scope of this problem.).
- The use of the term "sub-sampling" in this paper is ambiguous. In this paper, modality sub-sampling is a very important concept, and the authors show that it increases the irreducible discrepancy and makes the lower bound less tight. Sub-sampling here is supposed to mean extracting a subset from multiple subsets of modalities, but the authors do not define it clearly. Moreover, they also use the term to mean ELBO sub-sampling in MVAE, which may cause confusion. The authors should clarify the meaning of this term.

Minor comment:
- Is there any reason why the usual notation for encoders and decoders in VAEs ($p_{\theta}$ and $q_{\phi}$) are reversed in this paper? (Of course, there is nothing wrong with this, as it is consistent throughout the paper.)

**Summary Of The Paper:**

In this paper, the authors show theoretically that in mixture-based multimodal VAEs, sub-sampling of modalities can make ELBO less tight. They also introduced a more complex dataset, Translated-PolyMNIST and Caltech Birds with real images, to reveal the limitations of multimodal VAEs in practice.

**Summary Of The Review:**

Although some concerns remain about the terminology and the scope of the issue the authors are advocating, the paper makes some novel points and contributions to the study of multimodal VAEs. Therefore, I judge this paper to be above the acceptance.

---

> ### Author Response · Authors · 2021-11-19
> **reply to reviewer UGYa**
>
> > The authors define models with Equation 2 as the objective as mixture-based multimodal VAEs, and they show the limitations of Equation 2 due to sub-sampling. However, this objective is a lower bound of the "true" objectives of MMVAE and MoPoE-VAE, as pointed out by the authors. Therefore, it is unclear whether the problems pointed out in this paper occur only in Equation 2 or also in these true objectives (I understand that the actual implementation in [Sutter+ 2021] uses Equation 2, but my point is that the authors should clarify the scope of this problem.).
>
> Thank you for the excellent question. It is correct that the “true” objective of the MMVAE and MoPoE-VAE is the multimodal ELBO (Definition 1) and in principle both approaches can optimize a tighter bound via Equation 28 (in the revised version). Theoretically, the limitations apply to the tighter objective (Equation 28) in the same way as they do to Equation 2, because the discrepancy $\Delta(\bf{X}, \mathcal{S})$ stems from the likelihood term (and not from the KL-divergence). In particular, notice that the likelihood term is equal for Equation 28 and Equation 2 and that the lower bound (Equation 33) follows from a decomposition of the KL-divergence term. Therefore, our theory predicts the same discrepancy for the optimization of the “true” objective by the MMVAE and MoPoE-VAE. Our results from Figure 12 (described in Appendix C.3) confirm that the generative discrepancy can also be observed when we use an implementation that optimizes a tighter objective. Since the codebase from Sutter et al. (2021), that we use for our main experiments, implements the looser bound, we tend to keep the formulation of the limitations with regard to Equation 2, but to clarify the scope of the problem, we have extended the discussion (Section 6, first paragraph) to address your concerns.
>
> > The use of the term "sub-sampling" in this paper is ambiguous. In this paper, modality sub-sampling is a very important concept, and the authors show that it increases the irreducible discrepancy and makes the lower bound less tight. Sub-sampling here is supposed to mean extracting a subset from multiple subsets of modalities, but the authors do not define it clearly. Moreover, they also use the term to mean ELBO sub-sampling in MVAE, which may cause confusion. The authors should clarify the meaning of this term.
>
>
> Yes, sub-sampling refers to the extraction of subsets of modalities (from a set of subsets $\mathcal{S}$). While the term has not been used previously, it describes the computational procedure for the variational mixture posterior. To clarify the meaning of the term, we have extended the explanation of the computational procedure below Definition 3. To prevent confusion with the ELBO sub-sampling employed by (a version of) the MVAE, we have clarified in Section 2 (second paragraph) that the MVAE employs sub-sampling without reconstructing the modalities that are missing from the extracted subset.
>
> > Minor comment: Is there any reason why the usual notation for encoders and decoders in VAEs ($p_\theta$ and $q_\phi$) are reversed in this paper? (Of course, there is nothing wrong with this, as it is consistent throughout the paper.)
>
> Primarily, the notation is used for consistency with the referenced information-theoretic literature (Alemi et al, 2017; Poole et al. 2019), because we think that the information-theoretic perspective on VAEs leads to a more natural derivation of the generative discrepancy. Note that we discuss some notational differences in Appendix B.1.
>
> We hope that our response clarifies your concerns about the terminology and scope.

---

> > ### Comment · Reviewer_UGYa · 2021-12-01
> > **Reply to the authors**
> >
> > Thank you for addressing my concerns. Your reply and revision have addressed my concerns, so I will raise my rating.

---

### Official Review · Reviewer_tUe8 · 2021-11-02

**Correctness:** 4
**Technical Novelty And Significance:** 2
**Empirical Novelty And Significance:** 2
**Recommendation:** 6
**Confidence:** 3

**Main Review:**

Multi-modal data analysis is an important topic with several real-world applications. The authors' contribution is to characterize the limitations of existing VAEs as a step towards improving these methods, which is quite valuable. However, for this conference, I expect the authors to also suggest a solution for the discussed limitations. While it is quite useful to quantify the generative model performance in the presence of multi-modal data, it is not surprising to see that when all modalities are present, the joint distribution approximation for VAEs that use subsets of modalities (MMVAE) is not as accurate as VAEs that use the joint posterior as a product of unimodal posteriors (MVAE). These are discussed by Sutter et al., ICLR2021 as well.

The paper is very well written and well organized. Both theoretical and experimental results seem valuable.  However, it would be nice if the authors could show how we can overcome the lack of generative quality due to the discrepancy, $\Delta(x,s)$ in Theorem 1 and how we should balance the “sufficient diversity” and  “redundant information” with respect to the existing modalities. I think these are critical points that need to be somehow addressed.


**Summary Of The Paper:**

The manuscript attempts to study some of the existing multi-modal VAE models, MVAE, MMVAE, and MoPoEVAE and to identify their limitations in terms of generative quality and generative coherence. The authors study the impact of sub-sampling on the generative quality and show that if the shared information among modalities cannot be fully predicted across modalities, the model would lack generative coherence. They demonstrate their findings for two multi-modal datasets, i.e., PolyMNIST and Caltech Birds. To better show the limitations of multi-modal VAEs when applied to more complex data, the authors made an augmented version of PolyMNIST, called Translated-PolyMNIST including 5 modalities.

**Summary Of The Review:**

The authors' contribution on formulating and quantifying the limitation of the existing VAEs for multi-modal data analysis is interesting and valuable. However, I think the authors need to at least address a few of these limitations to show how the current methods can be improved.

---

> ### Author Response · Authors · 2021-11-19
> **reply to reviewer tUe8**
>
> > The authors' contribution is to characterize the limitations of existing VAEs as a step towards improving these methods, which is quite valuable. However, for this conference, I expect the authors to also suggest a solution for the discussed limitations.
>
> We appreciate that you recognize our work as a valuable step towards improving existing methods. While a solution would certainly be useful, we want to stress that our work identifies *fundamental* limitations of a large family of models. Our results point to the fact that these limitations cannot be resolved by straightforward extensions of the existing methods. Hence, to make the community aware of the fundamental problems, the goal of our work is to provide a comprehensive theoretical and empirical analysis of the limitations and their practical ramifications.
>
>
> > While it is quite useful to quantify the generative model performance in the presence of multi-modal data, it is not surprising to see that when all modalities are present, the joint distribution approximation for VAEs that use subsets of modalities (MMVAE) is not as accurate as VAEs that use the joint posterior as a product of unimodal posteriors (MVAE). These are discussed by Sutter et al., ICLR2021 as well.
> The insights of our work are markedly different compared to what was known from previous literature.
>
> While previous work (Sutter et al., 2021) observes a small discrepancy for a limiting case (when all modalities are present at test time), our paper establishes that this discrepancy is a symptom of a fundamental problem that lies at the heart of multimodal VAEs. Our work does not only quantify the generative performance in a large-scale study, but it provides a theoretical explanation of the problem and derives non-trivial implications for real-world applications. On  a more practical note: from our own experience we know that it is difficult to discern suitable use-cases for multimodal VAEs based on the claims and experiments in previous works. In particular, it is hard to understand what real-world datasets multimodal VAEs are suitable for, or what types of problems lie beyond their capability. To address this, we demonstrate the limitations of existing methods based on two simple failure cases that can help practitioners identify suitable use-cases. Hence, we are convinced that the understanding and scope of the limitations that are established in our work are not comparable with what was already known from previous literature.
>
> > The paper is very well written and well organized. Both theoretical and experimental results seem valuable. However, it would be nice if the authors could show how we can overcome the lack of generative quality due to the discrepancy, in Theorem 1 and how we should balance the “sufficient diversity” and “redundant information” with respect to the existing modalities. I think these are critical points that need to be somehow addressed.
>
> Thank you for recognizing the value of our theoretical and empirical results. Again, we want to stress that the focus of our work is to provide a comprehensive study of the fundamental limitations of existing methods and that our results suggest that these limitations cannot be resolved by straightforward extensions of these methods. For future work, we point out concrete ideas for potential solutions (and possible drawbacks thereof) in Section 6, but we are convinced that the implementation of a solution lies beyond the scope of this study. Regarding the second point, the information-theoretic concepts of diversity and redundant information, please note that these concepts do not stand in direct relation to a potential solution, because they describe properties of the data.
>
> We hope that our response helps you understand that our work deliberately focuses on the limitations and that the implementation of a solution lies beyond the scope of this work.

---

> > ### Comment · Reviewer_tUe8 · 2021-11-30
> > **Response to the rebuttal**
> >
> > I appreciate the authors' effort in addressing my questions. I have carefully read the rebuttal and would like to raise my score.

---

### Official Review · Reviewer_wNh5 · 2021-11-05

**Correctness:** 3
**Technical Novelty And Significance:** 3
**Empirical Novelty And Significance:** 3
**Recommendation:** 8
**Confidence:** 3

**Main Review:**

**Strengths**
* The authors do a great job covering a wide body of research on multi-modal VAEs.
* The paper proposes a theoretically grounded explanation for the empirically observed lower generation quality of multi-modal VAEs. The explanation appears sound.
* Extensive experiments on simple datasets are carried out to further support the results.
* Limitations of multi-modal VAEs discussed in the paper would be of interest of a wide audience beyond machine learning research.

**Weaknesses**
* The paper would benefit from a more detailed introduction of the datasets, especially PolyMNIST, e.g. why does it have 5 modalities and what are they?
* I was not fully convinced by the comparison between VAEs that rely on sub-sampling and those that do not. In particular, VAEs that rely on sub-sampling see less data during training compared to those that do not. Would conclusions change if VAEs that rely on sub-sampling would be trained proportionally longer?

**Summary Of The Paper:**

The paper discusses a surprising failure mode of multi-modal VAEs, in which their generation quality lags behind that of unimodal VAE. A theoretically grounded explanation based on the ELBO of the multi-modal VAE is proposed to explain this gap, and extensive ablation experiments validating implications of the proposed explanation are carried out. Finally, the theory and experiments proposed in the paper have implications for a large body of research into multi-modal VAEs.

**Summary Of The Review:**

Fundamental limitations of multi-modal VAEs discussed and experimentally validated in the paper would be of interest of a large audience and can potentially impact many applications of these methods (e.g. in natural sciences, bioinformatics, etc). Intuition and theoretical justification of the limitations appear sound, and is supported by extensive experiments.

---

> ### Author Response · Authors · 2021-11-19
> **reply to reviewer wNh5**
>
> Thank you for the positive feedback!
>
> > The paper would benefit from a more detailed introduction of the datasets, especially PolyMNIST, e.g. why does it have 5 modalities and what are they?
>
> Thank you for pointing this out. We have added a detailed description of the datasets to Section C.1.
>
> > I was not fully convinced by the comparison between VAEs that rely on sub-sampling and those that do not. In particular, VAEs that rely on sub-sampling see less data during training compared to those that do not. Would conclusions change if VAEs that rely on sub-sampling would be trained proportionally longer?
>
> We can confidently say that the results do not change, if we increase the training time of all multimodal VAEs. We already made sure that the models have converged, but to be absolutely sure, we trained individual models for 10 times longer and observed no significant changes in any one of the considered metrics.
>
> We hope that our reply clarifies the remaining concerns about the validity of the empirical claims. In this case, we would appreciate an adjustment of the “correctness” score.

---

### Author Response · Authors · 2021-11-19
**Reply to all reviewers**

Dear reviewers,

Thank you all for the constructive feedback!

We greatly appreciate the positive feedback from all of you with respect to the clarity and significance of our work. In particular, we appreciate that both our theoretical and empirical results were highlighted as novel and important by 3 out of 4 reviewers. While we agree with reviewer **tUe8** in that a solution to the presented limitations would be useful, the deliberate focus of our work is to provide a comprehensive study of the theoretical limitations and their practical ramifications. We appreciate that reviewer **tUe8** still recognizes that our work is an important step towards improving existing methods, but we believe that the implementation of a solution lies beyond the scope of this work.

In summary, we address the following concerns:

- Reviewer **wNh5**: we have checked that our conclusion remains unchanged, if we increase the training time for multimodal VAEs. Further, we have added a more detailed description of the datasets in Section C.1.
- Reviewer **tUe8**: our response makes clear that the focus of our work is to provide a comprehensive study of the limitations. We clarify how our results are markedly different from what was known from previous works.
- Reviewer **UGYa**: we have clarified the scope and terminology with minor changes to the manuscript. We have extended the discussion (Section 6) to explain that our theory predicts the same discrepancy for a tighter version of the objective. Further, we have clarified the use of the term “sub-sampling” in Section 2 and Section 3.2.
- Reviewer **B728**: we have clarified the relation between $\Delta$, the multimodal ELBO, and the empirically observed generative quality gap. Further, we have incorporated your suggestions about the possibility of an inductive bias (Sections 4.3 and 5.1) and have added definitions for the generative coherence metrics (Section C.2).

---

### Decision · Program_Chairs · 2022-01-20

**Decision:**

Accept (Poster)

**Comment:**

This paper provides an investigation into the quality of generations made by multimodal VAEs. All reviewers were in favor of accepting the paper, and there was quite a bit of detailed discussion and clarifications in the revised version of the paper which led two reviewers to raise their ratings. Overall this is an interesting contribution to the area and is an excellent fit for ICLR.